# Multi-model comparison of trends and controls of near-bed oxygen concentration on the Northwest European Continental Shelf under climate change

Giovanni Galli[1,4], Sarah Wakelin[2], James Harle[3], Jason Holt[2], Yuri Artioli[1]

[1]Plymouth Marine Laboratory (PML), Prospect Place, Plymouth, Devon, PL1 3DH, United Kingdom
[2]National Oceanography Centre (NOC), Joseph Proudman Building, 6 Brownlow Street, Liverpool, L3 5DA, United Kingdom
[3]National Oceanography Centre (NOC), European Way, Southampton, SO14 3ZH, United Kingdom
[4]National Institute of Oceanography and Experimental Geophysics (OGS), via Beirut 2, Trieste, 34014, Italy

*Correspondence to*: Yuri Artioli (yuti@pml.ac.uk)

**Abstract.** We present an analysis of the evolution of near-bed oxygen in the next century in the Northwest European Continental Shelf in a three-member ensemble of coupled physics-biogeochemistry models. The comparison between model results helps highlighting the biogeochemical mechanisms responsible for the observed deoxygenation trends and their response to climate drivers.

While all models predict a decrease in near bed oxygen proportional to climate change intensity, the response is spatially heterogeneous, with hotspots of oxygen decline (up to $-1$ mg L$^{-1}$) developing along the Norwegian Trench in the members with the most intense change, as well as areas where compensating mechanisms mitigate change.

We separate the components of oxygen change associated to the warming effect on oxygen solubility from those due to the effects of changes in transport and biological processes. We find that while warming is responsible for a mostly uniform decline throughout the shelf ($-0.30$ mg L$^{-1}$ averaged across ensemble members), changes in transport and biological processes account for the detected heterogeneity.

Hotspots of deoxygenation are associated with enhanced stratification that greatly reduces vertical transport. A major change in circulation in the North Sea is responsible for the onset of one such hotspot that develops along the Norwegian Trench and adjacent areas in the members characterised by intense climate change.

Conversely, relatively shallow and well mixed coastal areas like the Southern North Sea, Irish Sea and English Channel experience an increase in net primary production that partially mitigates oxygen decline in all members.

This work represents the first multi-model comparison addressing deoxygenation in the Northwest European Shelf and contributes to characterise the possible trajectories of near-bed oxygen and the processes that drive deoxygenation in this region.

As our downscaled members factor in riverine inputs and small and medium scale circulation which are not usually well represented in earth system models, results are relevant for the understanding of deoxygenation in coastal and shelf systems.

## 1 Introduction

Oxygen availability is of vital importance for aquatic life and the occurrence of low oxygen concentrations represents a major threat for marine (and freshwater) ecosystems. Exposure to oxygen levels below critical values do result in mortality, but also sub-lethal effects from reduced oxygen (e.g. depression of metabolism, Rubalcaba et al. 2020, impaired vision, McCormick et al. 2022) are of concern. Oxygen supply, which is modulated by temperature, has been proposed as a factor explaining maximal attainable size (Verberk et al., 2021) and geographic distribution (Deutsch et al., 2015) in marine species. Future deoxygenation, combined with warming, has hence the potential to alter marine ecosystems even where critical low oxygen concentrations are not exceeded. Bindoff et al. (2019) estimated that the world's marine oxygen inventory has been declining by 1.55±0.88% over the 1970-2010 period and a further decline of 3.45% compared to the 1990s is projected by the end of the century (Bopp et al., 2013) under the RCP8.5 scenario. A more recent estimate based on a CMIP5 and CMIP6 multi-model comparison estimated a global reduction of 9.51 to 13.27 mmol m$^{-3}$ (under RCP8.5 and SSP5-8.5 scenarios respectively) over the subsurface layer (100-600m) for the end of the century compared to pre-industrial values (Kwiatkowski et al., 2020).

Ocean oxygen concentration is determined by multiple complex processes. Air-sea gas exchange supplies oxygen to the upper ocean layers and primary production by phytoplankton contributes to a net production of oxygen in the euphotic layer, whilst away from the euphotic zone respiration removes oxygen. As organic matter (from marine primary and secondary production and of terrestrial origin) sinks, it is consumed by bacteria and zooplankton that consume oxygen in the process. Mixing and ocean currents are responsible for delivering oxygen to the deeper layers and where the ocean is stratified (either permanently or seasonally) the transport of oxygen from the surface to the bottom layer is inhibited. In addition to that, oxygen solubility is controlled by temperature (and salinity), hence ocean warming will reduce the amount of oxygen that can be dissolved in seawater.

The bottom layer of the oceans is especially vulnerable to oxygen depletion because it is isolated from the surface and frequently from the euphotic layer, and because sinking organic matter may accumulate there and be respired by bacteria and other organisms. In a similar vein, in the global ocean, oxygen minimum zones are generally found in subsurface waters where high surface productivity and associated high respiratory demand are accompanied by low oxygen supply due to sluggish circulation and weak vertical mixing (Oschlies et al., 2018). Near-bed oxygen concentration is a particularly significant indicator of ecosystem status because it dictates habitat viability for benthic sessile and scarcely motile species that cannot move quickly to more favourable conditions, and because benthic anoxic and hypoxic events are linked to eutrophication (Devlin et al. 2023).

The impact of climate change on oxygen change in coastal and shelf environments is at present not fully understood as it results from the interplay of multiple, often antagonistic, physical and biological processes that render ecosystem response highly uncertain and spatially heterogeneous. Due to their limited depth, shelf seas are projected to warm up more than the open ocean (Kwiatkowski et al., 2020). In addition, the coastal and shelf ecosystems are under stronger influence from

terrestrial nutrient and organic matter inputs, which can foster both primary production and respiration, while limited depth and tidal mixing favour mixing-induced oxygenation. Furthermore, the diagnosis of observed and predicted oxygen dynamics may depend on the spatial scales and domains being analysed: while Gilbert et al. (2010) found recent past median oxygen decline rates to be more severe in coastal waters (defined as a 30 km band near the coast) than in the open ocean (>100 km from the coast), Kwiatkowski et al. (2020) found benthic oxygen depletion not to be predominantly confined to shelf waters in a multi-model comparison. These two findings are not necessarily inconsistent as the definition of coastal waters, from Gilbert et al. (2010), differs from that of shelf waters from Kwiatkowski et al. (2020). However, it is important to note that the CMIP5 and CMIP6 earth system models used in the latter study are generally of coarse resolution (around 1 degree) and do not correctly resolve all coastal and shelf processes (e.g. small and medium scale circulation, riverine inputs), or coastlines and bathymetry.

The Northwest European continental shelf (NWES, Fig. 1) is located in the northeast Atlantic; it has an open connection with the Atlantic at its northern and western boundaries, along the continental slope, and with the Baltic Sea through the Skagerrak. A current runs northward along much of the continental slope and is responsible for exchange with the open ocean. Oceanic waters enter the North Sea north via the Fair Isle, East Shetland and Western Norwegian Trench currents and south via the English Channel; circulation in the North Sea is counter-clockwise with water exiting along the Norwegian Trench after having joined with Baltic Sea outflow. The NWES has relatively short flushing times of the order of 2-4y for the North Sea and 100d for the Norwegian Trench (Blaas et al. 2001). A detailed description of the North Sea physical oceanography can be found in Huthnance (1991) and Ricker and Stanev (2020). Much of the NWES stratifies seasonally, during the boreal summer, but relatively shallow coastal areas in the southern North Sea, Irish Sea and Western English Channel that are under strong influence from tides remain well mixed throughout the year. Here mixing maintains well oxygenated bottom waters year-round, while the Central North Sea, Celtic Sea, Armorican Shelf and Eastern English Channel are known to be prone to near-bed oxygen depletion (Breitburg et al., 2018; Ciavatta et al., 2016). In addition to that, marine ecosystems in the densely populated southern coastal regions of the NWES are currently facing multiple anthropogenic pressures (Korpinen et al. 2021).

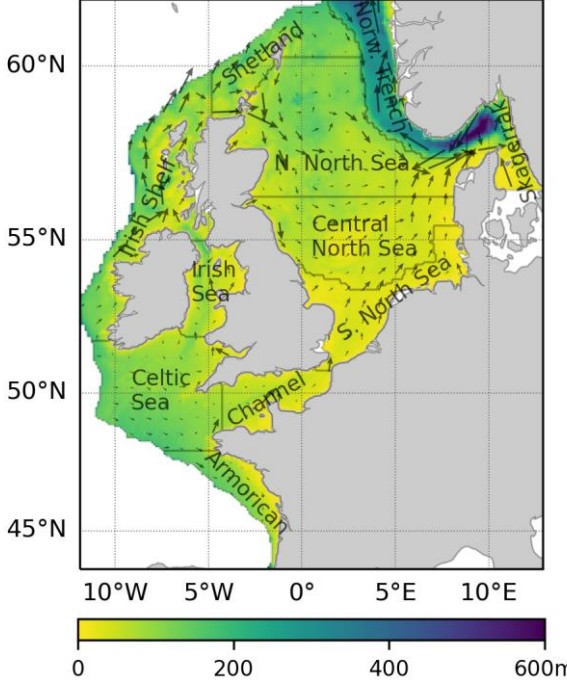

**Fig. 1.** The Northwest European Continental Shelf and its sub-basins considered in this study. Colour scale represents bathymetry. Arrows indicate the mean annual circulation for the sample year 2000.

When trying to assess the future evolution of oxygen in coastal and shelf ecosystems, given the uncertainty and spatio-temporal heterogeneity that characterizes projections of deoxygenation, coupled physics-biogeochemistry regional models are generally a more appropriate tool compared to global models (e.g. Holt et al. 2018, Fagundes et al. 2020, Markus Meier et al. 2021, but see Pozo Buil et al. 2021). The finer resolution of a bespoke regional model allows for a better representation of small to medium scale processes than would be possible with a coarser global model (Drenkard et al., 2021; Giorgi, 2019; Holt et al., 2016) and the implications of these processes for local climate impacts are at present understudied.

For the NWES, Wakelin et al. (2020) produced the first investigation on the potential change in near-bed oxygen during the 21st century under high greenhouse gas emissions (RCP8.5) using a downscaled climate projection. Wakelin et al. (2020) aimed at assessing not just the projected change in near-bed oxygen levels, but also at attributing such changes to driving physical and biogeochemical processes. Wakelin et al. (2020) found that while warming and freshening are generally coherent throughout the shelf, oxygen change displays pronounced spatial heterogeneity within sub-regions: the areas experiencing strong oxygen depletion become larger in the future and low oxygen periods last longer. This is due to the combined effect of warming, changes in transport and in biological processes, all components characterised by spatial heterogeneity. Wakelin et al. (2020) identify a large hotspot of oxygen depletion developing along the Norwegian Trench

and in the eastern part of the North Sea during the second half of the century where the major contribution to deoxygenation is increased ecosystem respiration, whereas the warming component is dominant elsewhere. This deoxygenation hotspot develops concurrently with a change-point in circulation that leads to reduced exchange between the Atlantic and the North Sea along its northern boundary (Holt et al., 2018).

Wakelin et al. (2020) is based on a single downscaled projection under a single climate change scenario (RCP8.5), hence it represents just one of the possible futures for the NWES. As the authors point out, the likelihood of the results can only be assessed through an ensemble of simulations that address multiple sources of model and scenario uncertainty.

Even within the same emission scenario, climate change projections display significant variability due to internal variability, e.g. phase of the climate system, and model variability, e.g. model structure and parameters, initial conditions, spin-up times, boundary conditions, etc., (Frölicher et al. 2016, Tapiador and Levizzani, 2021). In this study we built a small (three-members) ensemble of coupled physics-biogeochemistry regional models of the NWES all running for the 21st century under the RCP8.5 scenario. This ensemble is used to study the fate and controls of near-bed oxygen in the NWES. Each member was forced with lateral and atmospheric boundary conditions from one of three different earth system models from the CMIP5 collection (Taylor et al., 2012) that were chosen as they display a wide array of responses within the same emission scenario. One of our three members is the same used in Wakelin et al. (2020). Clearly a three-member ensemble is not sufficient to characterise all possible sources of uncertainty, nor to provide a robust assessment of the expected range of values. Instead, here we aim at investigating possible near-bed oxygen responses under a sufficiently large range of expected changes, while assessing how ecosystem processes change under different conditions, and whether certain processes are the same in all projections. Despite these limitations, such an exercise is useful to explore the uncertainty in projected near-bed oxygen change in the NWES, which is at present understudied, and, by identifying possible change trajectories, to provide a basis for future research. Our aim is to assess whether there's a coherent response in trends and controls across different members and climate change intensities, and whether the responses differ (or not) from those observed by Wakelin et al. (2020). Finally, we aim at clarifying whether the circulation change-point identified by Holt et al. (2018) drives near-bed oxygen change through similar mechanisms across models and change intensities, when the circulation change-point is present, and what happens when it is not.

## 2 Materials and methods

### 2.1 Ensemble description

All ensemble members use the NEMO-ERSEM model suite to downscale climate projections to the NWES domain and cover the 1980-2099 period under the RCP8.5 emission scenario, with a 10-year spin-up period (1980-1989) that was excluded from the present analysis. Spin-up times of the order of few years are the norm in NWES model runs (Tinker et al. 2014, Holt et al. 2018, Ciavatta et al. 2018) and are enough for the system to equilibrate; this is due to the short flushing times and highly dynamical nature of the NWES system, and indeed no significant drift was observed during the spin-up.

NEMO (Nucleus for European Modelling of the Ocean, Madec et al., 2019) is an ocean general circulation model and ERSEM (European Regional Seas Ecosystem Model, Butenschön et al., 2016) is a lower trophic network model that explicitly resolves the cycles of nutrients (N, P, Si), organic and inorganic carbon and oxygen in a coupled pelagic-benthic ecosystem. The downscaled domain covers 20°W to 13°E and 40°N to 65°N, including the NWES and adjacent deep ocean that was excluded from our analysis.

While two of the three members differ in boundary and initial conditions only, the third one, being an older simulation, also employs different NEMO-ERSEM code and vertical grid, it is hence not perfectly comparable to the other two. Commonalities and differences are detailed in the following.

Atmospheric and lateral oceanic boundary conditions were derived (in different ways, details below) from three Earth System Models (ESMs) from the CMIP5 collection (Taylor et al., 2012). Each ensemble member is forced with one separate

set of oceanic and atmospheric boundary conditions. The parent ESMs from which the boundary conditions are derived are GFDL-ESM2G (Dunne et al., 2012), IPSL-CM5A-MR (Dufresne et al., 2013) and HADGEM2-ES (Jones et al., 2011) and were chosen as they represent a gradient of climate sensitivities, with GFDL-ESM2G showing the lowest sensitivity and HADGEM2-ES the highest (Andrews et al., 2012). Simulation experiments yielded equilibrium climate sensitivities (global equilibrium surface-air-temperature change corresponding to an instantaneous doubling of atmospheric $CO_2$) of 4.59, 4.12

and 2.39K for HADGEM2-ES, IPSL-CM5A-MR and GFDL-ESM2G respectively (Andrews et al., 2012, Dufresne et al., 2013). The three downscaled ensemble members will be hereafter referred to as GFDL, IPSL and HADGEM for brevity. GFDL's and IPSL's oceanic boundary conditions were directly extracted from the parent ESM and interpolated onto the regional grid, while HADGEM uses as oceanic boundaries the output of a global coupled physics-biogeochemistry model (NEMO-MEDUSA, Yool et al., 2015) forced with atmospheric boundaries from the same HADGEM2-ES CMIP5 run.

All ensemble members have the same horizontal resolution of 1/15° latitude by 1/9° longitude (~7km) while the vertical resolution differs in HADGEM (33 vertical levels, s-coordinates, i.e. terrain-following) from the other two (51 vertical levels, s-coordinates). The NEMO version also differs in HADGEM (NEMO V3.2, O'Dea et al., 2012) from the other two members (NEMO V3.6, O'Dea et al., 2017); finally, the parameterization of ERSEM functional types is different in HADGEM (Blackford et al., 2004) with respect to the other two members (Butenschön et al., 2016). The reason for these

differences lies in the fact that HADGEM is an older run and was used in previous studies (Holt et al., 2018; Wakelin et al., 2020), while IPSL and GFDL were run more recently and are also part of a wider physics-only ensemble (see Holt et al 2022).

In IPSL and GFDL, physics initial conditions are from the parent ESM while biogeochemical variables are from a reanalysis product (Ciavatta et al., 2018), interpolated onto the regional grid. Ensemble members are forced with atmospheric

temperature, pressure, wind velocity, solar radiation, humidity and precipitation from the parent ESMs. Atmospheric nitrogen deposition is the same in the two members and comes from the EMEP project (Simpson et al. 2012, downloaded 2018). River discharge is from observed mean annual cycles for 250 rivers at daily frequency (Vörösmarty et al., 2000, Young and Holt, 2007), modulated by the fractional change (compared to 1984-2004 mean) in annual ESM precipitation

aggregated over four land regions (1. UK and Ireland; 2. Sweden and Norway; and Continental Europe: 3. east of 2.5°E and

4. west of 2.5°E). River nutrient loads are derived from the concentrations used in Ciavatta et al. (2018) multiplied by the discharge. Atmospheric $CO_2$ partial pressure (Riahi et al., 2007) is the same in the two members. To determine light availability for phytoplankton we forced the models with a climatological light attenuation coefficient field derived from CMEMS ocean colour products (Garnesson et al. 2021, level 3 product 009_086, marine.copernicus.eu).

IPSL's and GFDL's oceanic lateral boundary conditions were extracted from the parent ESMs and include all physics and

biogeochemistry variables (including oxygen); where biogeochemical variables were not present in the parent ESM (e.g. plankton functional types) they were set at close to zero values. IPSL's and GFDL's boundary conditions at the Baltic are instead forced with climatological mean values. Biogeochemical variables (including oxygen) were extracted from the World Ocean Atlas climatology (WOA, Boyer et al. 2018), physics form a reanalysis product (Kay et al. 2020), and interpolated onto the regional grid. Biogeochemical variables not present in the WOA dataset were set at close to zero

values. Finally, tidal forcing is as described in O'Dea et al. 2012 and O'Dea et al. 2017.

The HADGEM setup is similar, differing only in initial conditions (physics are from the NEMO-MEDUSA simulation that provides boundary conditions whilst biogeochemical variables are from a previous spun-up simulation), the version of EMEP nitrogen deposition (Simpson et al. 2012, downloaded 2011), the source of the light attenuation coefficient (Smyth et al., 2006) and the treatment of the Baltic boundary (freshwater inflows). HADGEM uses the same river forcing as IPSL and

GFDL but without modulation by the ESM precipitation. Also in HADGEM all biogeochemical oceanic boundary conditions, including oxygen, are set with a zero-gradient scheme, i.e. the concentration at the boundary equals the concentration immediately inside the domain. Full details of the HADGEM setup are given by Holt et al., (2018) and Wakelin et al., (2020).

Some validation of HADGEM can be found in Wakelin et al. (2020). A thorough validation of the NEMO-ERSEM

operational ecosystem model for the NWES can be found in Edwards et al. (2012) and in the Copernicus Quality User Information Document (Kay et al., 2020). Here we limit validation to the model variables that were considered in our analysis. Validation methods and results are presented in the supplementary material.

## 2.2 Analysis of oxygen change

Oxygen change can be partitioned into different components: a first one is related to warming that negatively affects oxygen

saturation concentration, hence lowering the amount of gas that can be effectively dissolved in seawater; another component is related to changes in biological processes that either consume or produce oxygen (respiration and primary production), and finally one component is related to change in transport processes responsible for oxygen supply (e.g. enhanced stratification limiting vertical transport and changes in circulation modifying lateral transport).

The Apparent Oxygen Utilisation metric (AOU, i.e. the difference between solubility and concentration) has traditionally been used as a measure of how much oxygen has been consumed (or produced) by biological processes since water has left

the surface (Duteil et al. 2013). AOU assumes that oxygen concentration is saturated ($O_2 = O_{2,sat}$) at the surface by virtue of ocean-atmosphere gas exchange, and that water temperature and salinity do not change after contact with the atmosphere. This way $O_{2,sat}$ of a water parcel does not change and any change in oxygen concentration, or equivalently in saturation state

(SS = $O_2 / O_{2,sat}$), is due only to transport (vertical mixing and lateral advection) and biological (primary production and respiration) processes. While it has been demonstrated that these assumptions are violated in the ocean interior and when undersaturated surface waters are subducted (Ito et al. 2004, Duteil et al. 2013), they still are a fair assumption in shallow and well mixed systems such as the NWES where vertical mixing dominates open ocean contributions.

Wakelin et al. (2020) proposed a variation of this method that focusses on the oxygen change at a given location with respect

to a reference period. Wakelin et al. (2020) partition oxygen change in two components, one related to change in $O_{2,sat}$ alone (i.e. to changes in temperature and salinity affecting oxygen solubility), the other related to change in SS, i.e. to transport (vertical mixing and lateral advection) and biological (primary production and respiration) processes. Here we present and employ a slightly revised version of Wakelin et al. (2020) method. The differences are minimal and the influence on the results was found to be, at least for our case study, negligible. With t0 being the reference time and t any subsequent time,

under the assumption that $O_{2,sat}$ and SS are independent, the partitioning of oxygen change between t0 and t can then be expressed as follows:

$$\Delta O_2 = \Delta(O_{2,sat}SS) = (SS_{t0}\Delta O_{2,sat}) + (O_{2,sat,t0}\Delta SS) + (\Delta SS\Delta O_{2,sat}) \tag{1}$$

The first term captures oxygen changes that are related to changes in $O_{2,sat}$ alone ($O_{2,sat,t}$ being the only non-constant term), hence to how temperature and salinity affect oxygen solubility, we will refer to this term as $\Delta O_{2,phy\text{-}ch}$ (physico-chemical). The second term captures changes that are related to changes in SS ($SS_t$ being the only non-constant term), hence to changes in transport and biological processes, we will refer to this term as $\Delta O_{2,other}$. The third term is a second order term and is related to both changes in $O_{2,sat}$ and SS, we will refer to this term as $\Delta O_{2,sord}$.


$$\Delta O_{2,phy-ch} = SS_{t0}\Delta O_{2,sat} \tag{2}$$
$$\Delta O_{2,other} = O_{2,sat,t0}\Delta SS \tag{3}$$
$$\Delta O_{2,sord} = \Delta SS\Delta O_{2,sat} \tag{4}$$

The metrics in eq. 2, 3, 4 were computed on a monthly basis for all three ensemble members and averaged over one climatological 30 years period to remove the effect of inter-annual variability. We chose as reference period t0 are monthly climatological values for the first 30y (1990-2019) and a similar period at the end of the century was selected to assess the impact of climate change on oxygen (2070-2099). $O_{2,sat}$ was computed from temperature and salinity using the relation described in Weiss (1970). SS was computed as $O_2 / O_{2,sat}$.

Eq. 2 and 3 show that, for the purpose of calculating correlations, the signal of the component of change related to change in solubility is captured by $O_{2,sat}$, while SS captures the signal related to all other processes. The metrics in eq. 2 and 3 are related to the classic $O_2 = O_{2,sat} - AOU$ decomposition. A detailed description of commonalities and differences of the two approaches can be found in the supplementary material, where it is shown that at the temporal and spatial scale used in this study the difference is small.

To assess what drives the oxygen changes, we computed, at each grid point, the correlations (spearman correlation, R, with significance threshold p<0.01) between monthly averaged timeseries (for the whole simulation) of $O_{2,sat}$ and SS and a number of physical and biogeochemical variables for which causal links could be plausible within the model's architecture. We decided to compute correlations between $O_{2,sat,t}$, $SS_t$ (instead of $\Delta O_{2,phy-ch}$, $\Delta O_{2,other}$) and other variables because, due to the definitions in eq. 2 and 3, correlations would not change. Due to system complexity and interconnection of processes, it

is expected for correlations to vary in space and time, with patterns that may not be straightforward to explain with simple direct causal links. If significant correlations were identified between variables that were not connected by a causal link able to justify that correlation, we looked at possible covariances between those and other variables that could explain the correlation. To investigate physical oxygen controls we considered atmospheric and near-bed temperature, surface salinity and potential energy anomaly (PEA, an indicator of stratification, de Boer et al. 2008). To investigate biogeochemical

oxygen controls we looked at depth-integrated net primary production and near-bed community and bacterial respiration. To assess how the change in Western Norwegian Trench current flux (Holt et al. 2018) influences oxygen trajectories, we compared the intensity of the current with near-bed $O_{2,sat}$ and SS values averaged over the Norwegian Trench Region. Average seasonal (monthly) cycle was removed from the time-series prior to the calculation of correlation coefficients to minimise type I errors (i.e. false positives) when the seasonal cycle dominates the correlation (Legendre and Legendre,

2012). When long-term trends are present, pairs of timeseries may be significantly correlated even in the absence of direct causal links (false positives). This can happen when the trend dominates the correlation. Detrending the timeseries prior to the calculation of correlation coefficients is an option. However, trends should be retained if they are the object of the analysis (Legendre and Legendre, 2012), as it is in this case. If trends are retained additional care must be taken in identifying all possible covariances that offer alternative explanations for the detected correlations. Here we did retain long-

term trends. An exploratory analysis that did use detrending (not shown), only found a slight degradation of detected correlations, and no relevant changes in sign.

     Hotspots of oxygen decline, where present, were identified as areas with $\Delta O_2 < -0.5$ mg $L^{-1}$ and $\Delta O_2 < 1.5 \Delta O_{2,mean}$ with $\Delta O_{2,mean}$ being the shelf average $\Delta O_2$.

When it comes to the response to oxygen concentrations of aquatic animals, absolute low oxygen thresholds are more meaningful than relative change. We addressed this by computing the incidence of hypoxic events in our three members under present and future conditions. Hypoxia incidence is calculated as the fraction of each present and future 30y period with near-bed oxygen falling below the threshold of 6 mg $L^{-1}$ that has been indicated as meaningful for the North Sea

ecosystems (OSPAR, 2003). Since here we look at absolute values, rather than relative change, model bias matters, therefore we bias-corrected our models by subtracting the difference between model present day climatology and the North Sea Biogeochemical Climatology (NSBC) dataset (Hinrichs et al., 2017).

## 3 Results

In this section we will first summarise the impacts of climate change on temperature and salinity to provide some context on the main relevant climatic trends in the NWES (3.1), then we will look at the changes of near bed oxygen (3.2), and which of the two components of the decomposition is most contributing to those changes (3.3). The following two sections are devoted to investigating the local controls of the projected changes, in particular temperature and stratification (3.4), and biological processes (3.5). Finally, section 3.6 will analyse the role that changes in circulation in the North Sea have in driving those local controls.

### 3.1 Changes in temperature and salinity

All three models consistently predict warming and freshening of the NWES (Fig. 2, S3, S4). The climate sensitivity ranking of the three parent ESMs (Andrews et al., 2012) is reflected in the downscaled projections, with HADGEM showing the highest warming and freshening, GFDL the lowest and IPSL in between. The projected surface warming is mostly uniform throughout the NWES, with slightly more intense warming in shallower areas, whereas freshening is more intense along the Skagerrak and Norwegian Trench (also in GFDL where freshening is however very low) and in the eastern portion of the North Sea.

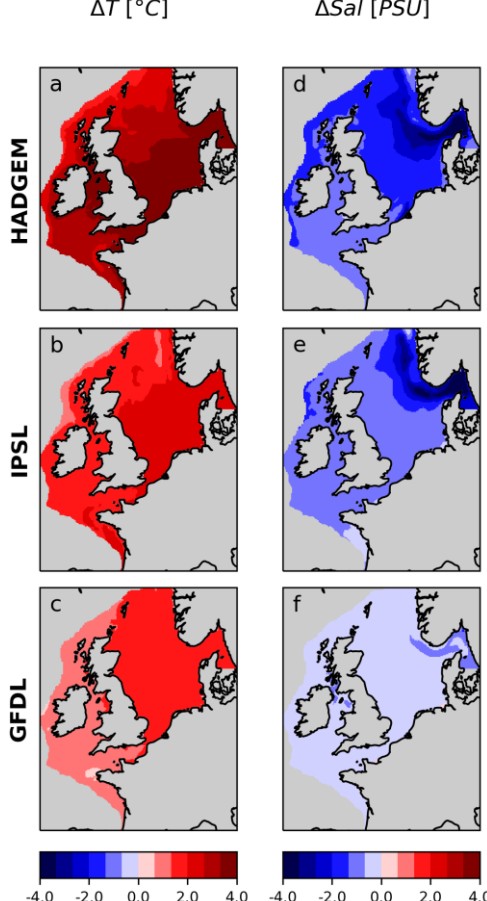

**Fig. 2.** Surface temperature and salinity, difference between future (2070-2099) and present (1990-2019) periods.

### 3.2 Near-bed oxygen current state and change

All three ensemble members consistently show a decrease in near-bed oxygen throughout the shelf (Fig. 3d, e, f, Fig. S5).

The severity of the impacts follows the three members' climate sensitivity: $-0.52$, $-0.36$ and $-0.14$ mg L$^{-1}$ for HADGEM, IPSL and GFDL respectively, averaged over the shelf. GFDL shows no relevant hotspots of oxygen decline; IPSL shows localized hotspots of intense oxygen decline along the Skagerrak and Norwegian Trench ($\sim-0.92$ mg L$^{-1}$ for Skagerrak and Norwegian trench combined) and along the western shelf margin (hatched areas in Fig 3d) but less severe impacts, similarly to GFDL, on the rest of the shelf; HADGEM shows the severest impacts, with a mean decline of $-0.74$ mg L$^{-1}$ over

Norwegian Trench, Skagerrak, Northern and Central North Sea combined, while also in the rest of the domain near-bed oxygen declines more than in the other two members.

Intense change in HADGEM results in widespread exceedance of hypoxia thresholds (Fig. 4g-l) in the North Sea under future conditions; this feature is still present but less prominent in IPSL and even less so in GFDL. Interestingly, in IPSL and GFDL areas of hypoxia occurrence in the Central North Sea coincide with the highest present $O_2$ levels.

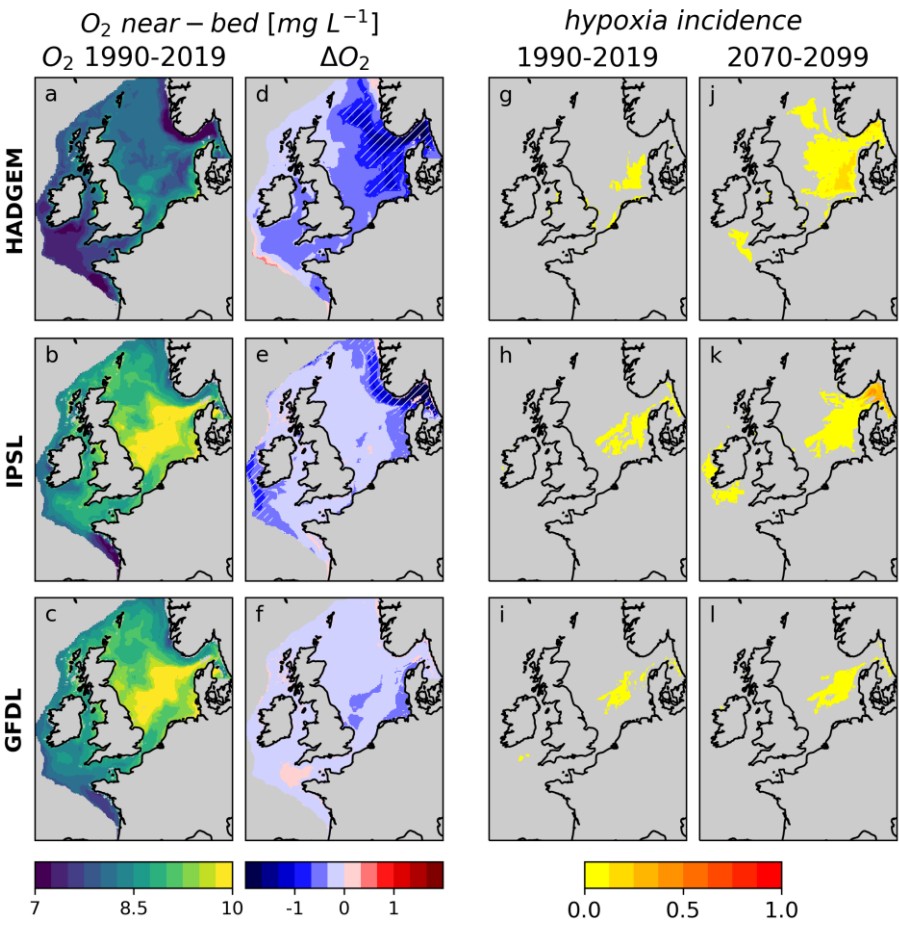


**Fig. 3.** Near-bed $O_2$ concentration, present state (average of 1990-2019) and change (difference of the 2070-2099 and 1990-2019 averages) and fraction of year with average near-bed Oxygen < 6 mg $L^{-1}$ under present day and future conditions, calculated on bias-corrected data. Hatched regions in d, e, f are deoxygenation hotspots.

### 3.3 Contributions to near-bed oxygen change

Figure 4a-f shows how the components of oxygen change ($\Delta O_{2,\text{phy-ch}}$ and $\Delta O_{2,\text{other}}$) contribute to the total change projected by the models (shelf-averaged $\Delta O_{2,\text{sord}}$ is < 1% of total change in all members, not shown). $\Delta O_{2,\text{phy-ch}}$ is fairly uniform throughout the shelf and its intensity follows the climate sensitivity of the driving ESM ($-0.38$, $-0.30$ and $-0.22$ mg $L^{-1}$ in HADGEM, IPSL and GFDL respectively, averaged over the shelf, $-0.3$ mg $L^{-1}$ averaged over all members). $\Delta O_{2,\text{other}}$ instead

shows significant variability both across models and, spatially, within models, with large areas even showing an increase. In

GFDL $\Delta O_{2,other}$ accounts for ~+0.1 mg $L^{-1}$ throughout the shelf, partially counterbalancing $\Delta O_{2,phy-ch}$ (~−0.22 mg $L^{-1}$). In IPSL $\Delta O_{2,other}$ is strongly negative in the Norwegian Trench and Skagerrak (−0.67 mg $L^{-1}$) and, to a lesser extent, along the western shelf margin, thus explaining the hotspots of oxygen decline in these areas, while $\Delta O_{2,other}$ is positive in the North Sea (~+0.08 mg $L^{-1}$ over all the North Sea), partially counterbalancing $\Delta O_{2,phy-ch}$ (−0.36 mg $L^{-1}$). In HADGEM the vast hotspot of declining near-bed oxygen in the eastern part of the North Sea, Norwegian Trench and Skagerrak is explained by

the combined effect of $\Delta O_{2,phy-ch}$ and $\Delta O_{2,other}$, with the latter accounting for the largest share of the decline (mean $\Delta O_{2,phy-ch}$, $\Delta O_{2,other}$ are −0.36 and −0.55 mg $L^{-1}$ in the Skagerrak and Norwegian Trench combined), while in the western North Sea, English Channel, Irish Sea and western shelf margin $\Delta O_{2,other}$ is positive (+0.12 mg $L^{-1}$ on average over Southern North Sea, Channel and Irish Sea). Fig. 4g-l also shows how $\Delta O_{2,sat}$ closely traces $\Delta O_{2,phy-ch}$ and $\Delta SS$ closely traces $\Delta O_{2,other}$, as expected from eq. 2 and 3. Maps showing present and future $O_{2,sat}$ and SS can be found in the supplementary material, Fig. S6, S7.


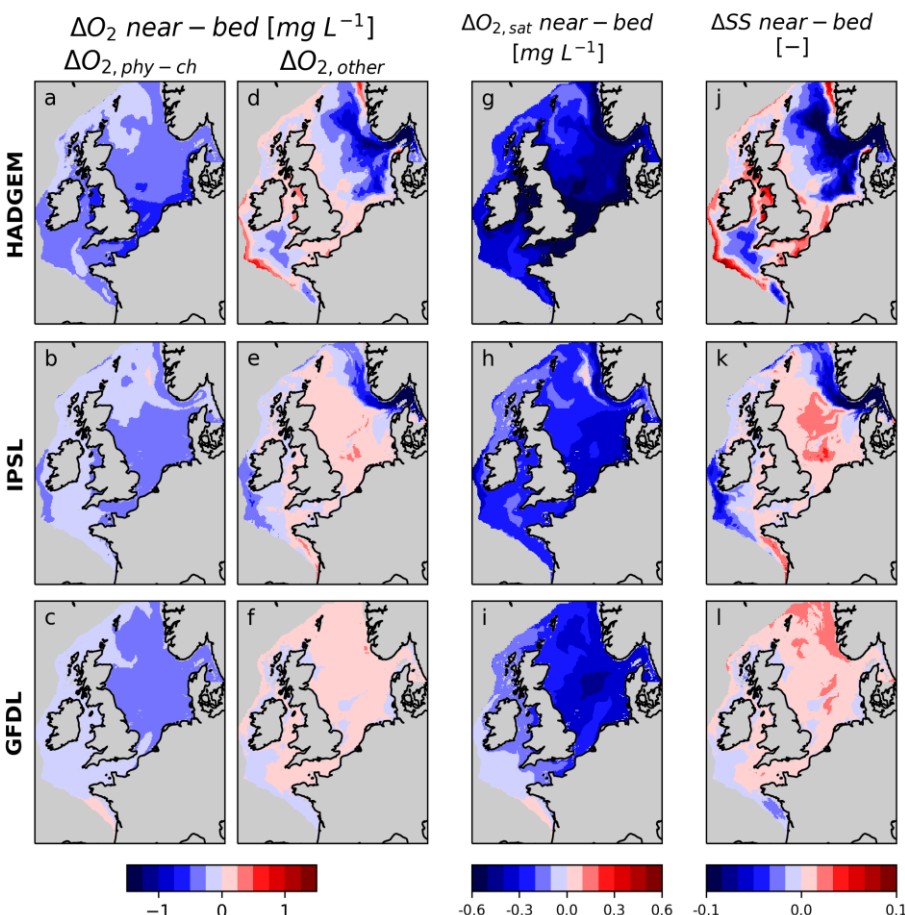

**Fig. 4.** Contributions of near-bed oxygen change, $\Delta O_{2,\text{phy-ch}}$ and $\Delta O_{2,\text{other}}$, and changes in SS and $O_{2,\text{sat}}$. Note that the spatial distribution of $\Delta O_{2,\text{sat}}$ and $\Delta SS$ closely follows that of $\Delta O_{2,\text{phy-ch}}$ and $\Delta O_{2,\text{other}}$ respectively.

**3.4 Physical controls of oxygen change: temperature and stratification**

Changes in $O_{2,\text{phy-ch}}$ and $O_{2,\text{sat}}$ are, for the greatest part, explained by warming (correlation between $O_{2,\text{sat}}$ and near-bed T $\sim$−1 everywhere in all models, not shown). The driver of this is the temperature atmospheric forcing ($T_{\text{atm}}$, Fig. 5a, b, c) that is negatively correlated with near-bed $O_{2,\text{sat}}$ throughout the domain in all members. $T_{\text{atm}}$ also correlates with SS with heterogeneous patterns throughout the shelf (fig. 5d, e, f). The correlation is negative along hotspots of oxygen decline in the Skagerrak and Norwegian Trench in IPSL and HADGEM (and the eastern part of the North Sea in HADGEM only). The correlation is instead positive in the Southern North Sea, Channel and Irish Sea (all members) and in the Northern and Central North Sea (IPSL and GFDL). While $T_{\text{atm}}$ and SS covary, we do not interpret these correlations as a direct causal link. Change in SS in these areas is better explained by different processes, as will be explained in the following sections.

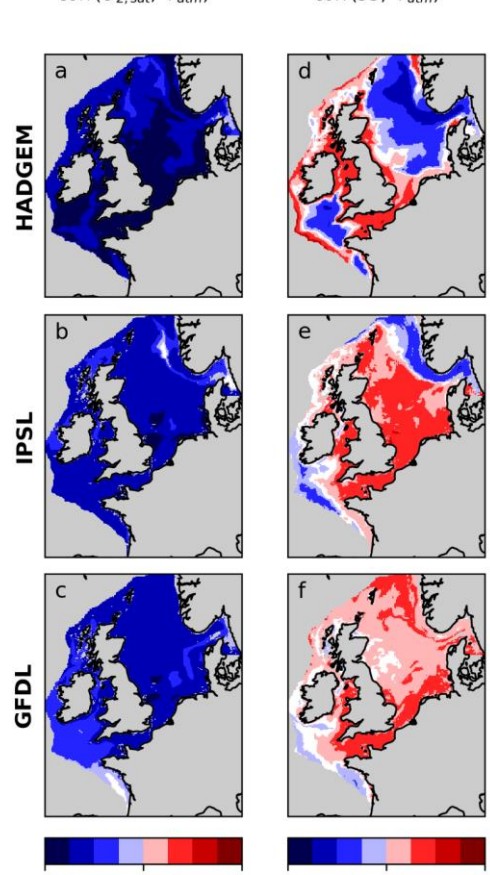

**Fig. 5.** Correlation between temperature atmospheric forcing and near-bed $O_{2,sat}$ and SS. White areas on the shelf indicate non-significant correlation.

The hotspots of oxygen decline in HADGEM and IPSL coincide with enhanced stratification hotspots (red areas in Fig. 6a, b). SS and PEA are, in both ensemble members, strongly negatively correlated in these areas (Fig. 6g, h). GFDL on the other hand only shows a moderate increase in stratification and no significant hotspots, with a weaker correlation between SS and PEA. In all members the main driver of stratification along the Norwegian Trench and in the eastern part of the North Sea is surface salinity, that is negatively correlated with PEA, in all members (Fig. 6d, e, f). The positive correlation between PEA

and SS in coastal areas in the Southern North Sea and around the British Isles (observed in all members) appears to be caused by the seasonality of primary productivity. These shallow regions experience strong tides and remain well mixed year-round (PEA barely changes in the long term). Here stratification is not a meaningful indicator of vertical oxygen transport. However, the highest PEA values do happen in the summer months, when also NPP peaks, producing oxygen that contributes to high SS values, while the opposite is true during winter; hence the positive correlation (see 3.5).

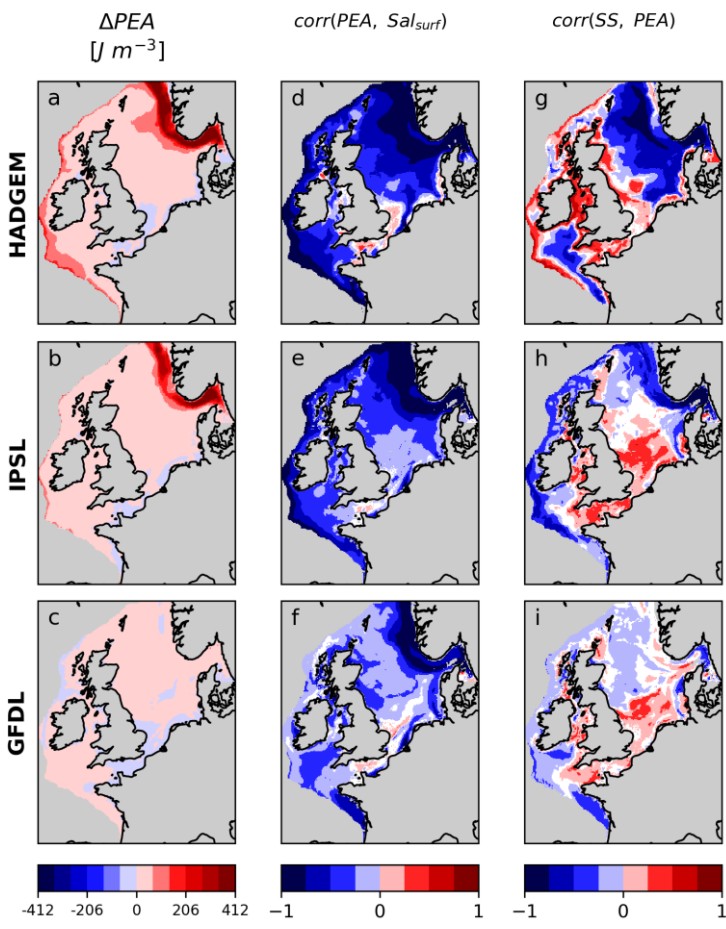

**Fig. 6.** Change in potential energy anomaly (PEA) and correlation between PEA and surface salinity and PEA and SS. White areas on the shelf indicate non-significant correlation.

### 3.5 Biogeochemical controls of oxygen change: primary production and respiration

In HADGEM depth integrated net primary production (NPP) increases over much of the North Sea, with the exception of the northwestern sector (Fig. 7a). In IPSL and GFDL instead NPP decreases in the North Sea and along the shelf edge due to decreasing oceanic nutrient input (Fig 7b, c), similarly to what was shown by Holt et al. (2012, 2016). In all members, in the shallow and well mixed southern North Sea, English Channel and Irish Sea NPP increases providing additional oxygenation, as shown by the positive correlation between SS and NPP (Fig 7d, e, f).

In HADGEM NPP is negatively correlated with SS in the hotspot of oxygen decline in the eastern part of the North Sea, Skagerrak and Norwegian Trench (Fig. 7d). In these deeper and seasonally stratified areas, increasing NPP contributes to oxygen decline by producing organic matter that sinks and is later respired (see below). This also happens in IPSL but limited to the Skagerrak (Fig. 7e).

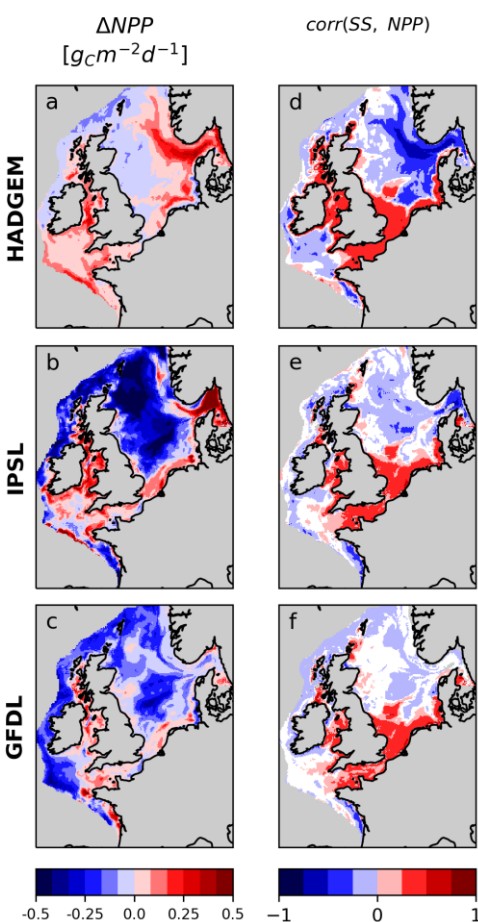

**Fig. 7.** Change in depth integrated net primary production (NPP) and correlation between NPP and SS. White areas on the shelf indicate non-significant correlation.

Bacterial respiration is the largest contribution of total community respiration, given the faster turnover, hence here we will focus only on this component. Results do not change significantly when community respiration is considered (not shown).

In HADGEM near-bed bacterial respiration (BResp) increases in the eastern part of the North Sea (Fig. 8a), fuelled by increasing NPP, thus contributing, in tandem with enhanced stratification, to oxygen decline; SS and BResp are indeed significantly negatively correlated in this area (Fig. 8d, g).

In IPSL and GFDL instead BResp decreases throughout most of the North Sea and along the shelf margin (Fig. 8b, c), suggesting reduced oxygen consumption as a cause for the observed increase in SS. However, when all monthly values are considered, the correlation between SS and BResp is rather weak and, at some locations, positive (instead of negative as would be expected, Fig. 8e, f). This is because simple point-to-point correlation over the full period, does not allow to capture seasonally heterogeneous processes. Conversely, a significant negative correlation between SS and BResp is

detected for the Central and Northern North Sea in both IPSL and GFDL when singling out the months from November to March (Fig. 8h, i). This is because during winter months, with little primary production, respiration is a dominant contribution to oxygen levels. Instead, during the remainder of the year (growth season), The correlation is weakly positive or non-significant (not shown).

In IPSL along the Norwegian Trench BResp and SS are positively correlated, due to both variables decreasing. Here the driver of the decrease in BResp is the decrease in NPP along the trench, while the decline in SS is related to increasing stratification (see section 3.4). While the two variables covary there doesn't seem to be a strong direct causal link.

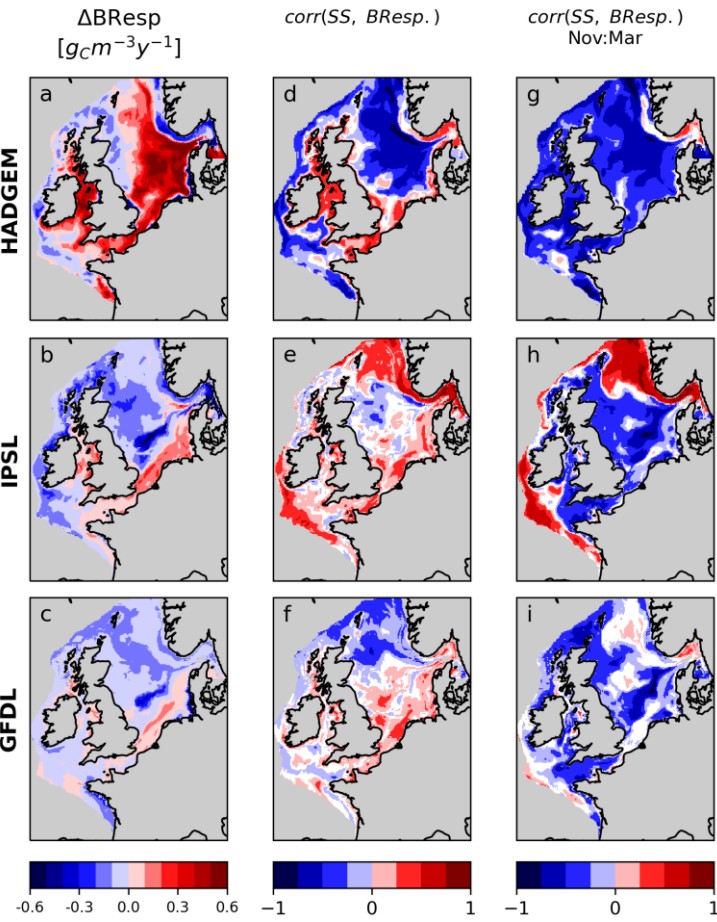

**Fig. 8.** Change in near-bed BResp and correlation between BResp and SS for all months and for months from November to March alone. White areas on the shelf indicate non-significant correlation.

**3.6 Impact of abrupt changes in circulation on the emergence of de-oxygenation hotspots**

425 The onset of the development of deoxygenation hotspots in the North Sea, Skagerrak and Norwegian Trench in HADGEM and IPSL is tied to a progressive weakening and reversal of the western Norwegian Trench current (wnt, Fig. 9a,b) starting approximately in the mid 2020s for both models (Holt et al., 2018). The time evolution of SS in this area is tightly coupled with that of the western Norwegian Trench current in both members (R, p = 0.77, 0.0 for HADGEM, 0.94, 0.0 for IPSL). This circulation change is absent from GFDL (Fig. 9c) that also lacks significant deoxygenation hotspots and correlation

430 between SS and western Norwegian Trench Current (R, p = 0.08, 0.0).

This suggests that the circulation change, by driving the observed freshening and increase in stratification in the North Sea, is the main driver of the development of deoxygenation hotspots. The time evolution of $O_{2,sat}$ is also coupled with current flux in HADGEM and IPSL (R, p = 0.89, 0.0 for HADGEM, 0.81, 0.0 for IPSL) and not as much in GFDL (R, p = 0.16, 0.0). This though cannot explain the deoxygenation hotspot as the change in $O_{2,sat}$ is homogeneous throughout the shelf (Fig.

435 4).

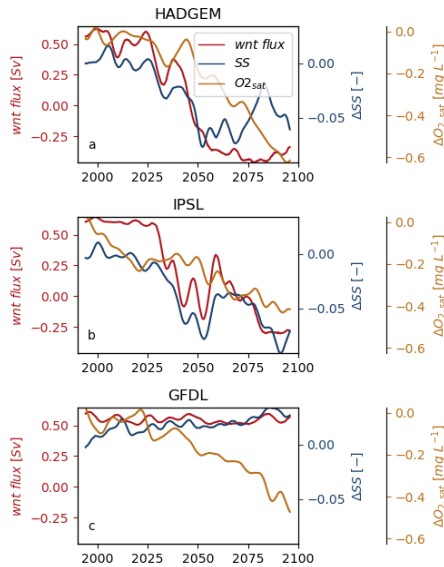

**Fig. 9.** Temporal evolution of $\Delta SS$ and $\Delta O_{2,sat}$ in the Norwegian Trench and Western Norwegian Trench current flux; monthly average data are smoothed with a Gaussian filter. Positive values of the current are entering the North Sea.

**4 Discussion**

440 We studied the spatio-temporal evolution of near-bed oxygen concentration in the NWES in a three-member ensemble of coupled physics-biogeochemistry downscaled climate projections running from 1980 to 2100 under a high emission scenario (RCP8.5). Building on previous work on oxygen (Wakelin et al., 2020) and circulation changes (Holt et al., 2018) in the

NWES, we investigated a wider range of projected change by using three instances of the same model suite (albeit with one member differing in model version and parameterisation) forced with boundary conditions from three global models covering a wide spectrum of climate sensitivities. This allowed to verify whether the members' response, and the processes involved, show any commonalities and/or differences, and how ecosystem response is related to the projected intensity of climate change.

In agreement with global models results (Kwiatkowski et al., 2020), all ensemble members consistently predicted a decline in near-bed oxygen throughout the shelf. This may contribute exacerbating ecosystem impacts in regions, such as the North Sea, which are already heavily impacted by multiple anthropogenic stressors like trawling, eutrophication, hazardous substances, noise, etc. (Korpinen et al. 2021). Our results confirm those of Wakelin et al. (2020), that, whilst responses of the physical system (warming, freshening) are generally homogeneous throughout the shelf, near-bed oxygen change can display marked spatial heterogeneity with hotspots of change, in this case related to circulation changes, as well as areas where antagonistic processes (decreasing respiration and, in well mixed regions, increasing NPP) mitigate oxygen decline. This spatial heterogeneity reflects changes in circulation, vertical transport and biological processes, which can contribute with changes of either sign to oxygen trends. On the contrary, warming-driven change in solubility produces a more homogeneous negative contribution to oxygen decline. This differential contribution to components of projected oxygen change has been observed also in global models (Kwiatkowski et al. 2020). It is well known that current climate models tend to underestimate recent rates of oxygen decline (Oschlies et al., 2018, 2017). However, some authors pointed out how modelled solubility-driven changes largely agree with observations, hinting at model deficiencies in representing biogeochemical cycles and changes in circulation and mixing processes (Oschlies et al., 2018). This is especially true for coastal and shelf ecosystems such as the NWES, where small scale processes are poorly represented in global models. This includes not just small and medium scale circulation, but also the level of detail with which biogeochemistry is represented, which is fairly simple in several global models (Kearney et al., 2021). Here we have shown how circulation and ecosystem processes can indeed account for a large portion of projected near-bed oxygen change in the NWES.

Compared to Wakelin et al. (2020), by using an ensemble rather than a single run, we show that hotspots of oxygen decline occur only in the members with the severest change (HADGEM and IPSL), whilst for relatively low climate change (GFDL) oxygen decline is largely homogeneous, albeit still negative. The large areas experiencing hypoxia ($O_2 < 6$ mg L$^{-1}$) identified in Wakelin et al. (2020) only emerge when the strongest climate change is projected by HADGEM, whilst the other two members are largely spared. Interestingly, in IPSL and GFDL areas experiencing (limited) hypoxia in the Central North Sea coincide with the highest present $O_2$ levels. These are highly productive areas (hence the high $O_2$ production) that stratify seasonally (hence vulnerable to oxygen depletion). Since our main focus here is on change in average values rather than extremes, we didn't further investigate this. However, it is interesting to note how these areas do not coincide with the highest $\Delta O_2$. This suggests that the dynamics of average and extreme values may not necessarily be coupled. Whereas Wakelin et al. (2020) pointed at near-bed bacterial respiration, fuelled by increased surface productivity, as the main driver of the emergence of deoxygenation hotspots, we highlight here how also enhanced stratification must be present to result in

significant declines in near-bed oxygen concentration. Indeed, in IPSL a deoxygenation hotspot develops along the Norwegian Trench also in the absence of an increase in near-bed respiration. Weak vertical mixing is indeed a characteristic of oxygen minimum zones in the global ocean (Oschlies et al. 2018). Although the deoxygenation hotspots we detect are on a much smaller scale, the processes involved in their formation are similar.

Our results also highlight the importance of circulation changes in driving deoxygenation processes in the NWES. In the two most climate sensitive ensemble members the onset and development of deoxygenation hotspots is tightly coupled with a major circulation change in the area that largely limits ocean-shelf exchange processes along the northern boundary of the North Sea. This circulation change has already been identified by Holt et al. (2018) by using the same model run as our HADGEM member; according to the authors, this decrease in western Norwegian Trench inflow can be traced to a substantial increase in stratification at the northern entrance to the trench, limiting the ability of the slope current to steer into the North Sea. This triggers a feedback mechanism where reduced exchange with the open ocean contributes to a freshening of the North Sea by increasing retention times of fresher water from continental Europe and the Baltic, thus driving a further increase in stratification. As the circulation change is observed only in the two members with the highest change levels, the results here are consistent with the observation by Tinker et al. (2016) that the large circulation changes occur only in downscaled projections with high climate sensitivity in the driving climate model (3 out of 11 ensemble members in that case). It is also worth noting that in Holt et al. (2018) this circulation change only emerges in the downscaled projection (the HADGEM simulation used here) and is not captured in the global model used to force the downscaling. Although we did not assess if this is the case also in our other two ensemble members, it appears likely as at the 1deg resolution of the parent ESMs the bathymetry and Baltic exchange will be poorly represented (in HADGEM2-ES the Baltic is closed altogether), and the slope current which turns into the North Sea will be far too diffuse because the slope is too broad.

Our ensemble uses climatological boundary conditions at the Baltic, so any changes that may happen in the Baltic are not accounted for in our analysis. Despite this limitation this choice also allows to rule out lateral transport of oxygen poor Baltic water as a factor contributing to deoxygenation hotspots in the ensemble.

As mentioned earlier, the model set-up and downscaling methods used in HADGEM differ from those in IPSL and GFDL (notably in vertical resolution, biogeochemistry parameterisation and boundary condition scheme), providing uncontrolled degrees of freedom to our multi-model comparison. This does not seem to have first order consequences for ocean physics, as the response of the three models in terms of changes in temperature, salinity and stratification appears largely coherent with the progressive levels of climate sensitivity represented. Furthermore, the biogeochemical response in the Southern North Sea, English Channel and Irish Sea is largely comparable across models. However, in the eastern part of the North Sea and Norwegian Trench HADGEM displays a noticeably different response when it comes to changes in primary production and respiration and in the correlation between these variables and near-bed oxygen. In particular, the link between net primary production, respiration and near-bed oxygen appears much tighter in HADGEM, as testified by stronger correlations. This is likely a consequence of the change in the parameter set of the biogeochemical model for both phytoplankton and bacteria in the recent update (Butenschön et al., 2016) compared to the original one (Blackford et al.,

2004), although also the different vertical discretisation may play a role here. Disentangling the exact causes of these differences is not trivial and would require a set of ad-hoc experiments which are out of the scope of this study. Despite this, the trends and drivers are still largely coherent across ensemble members (e.g. both HADGEM and IPSL develop a deoxygenation hotspot along the Norwegian Trench), testifying how our results are robust with respect to the model uncertainty represented.


While our results provide useful information about projected near-bed oxygen change in the NWES, our small ensemble doesn't sample variability (Frölicher et al. 2016) adequately enough to provide a robust estimate of change or associated uncertainty. In particular, this study does not address internal variability, nor it does address scenario variability as RCP8.5 only is used, and only partially addresses model variability through different forcings and model versions.


Here we mostly analysed model output by mapping long-term trends and point-to-point correlations between variables. Whilst this approach has proven useful in highlighting potential cause and effect mechanisms, it also has limits. Such an approach is prone to failure in identifying significant correlations when transport makes causes and effects spatially decoupled, for example in highly advective systems such as the Norwegian Trench, and/or when the relations between system variables change in time, either seasonally or on multi-annual timescales. Our method, similarly to other metrics like


Apparent Oxygen Utilisation (AOU), also assumes that biology and transport are the sole contributors to deviation from saturation at any point in space and time. This may not be true in subduction regions where undersaturated surface water is subducted and in the presence of sea ice (Duteil et al. 2013), or if the temperature and/or salinity of a water mass changes away from the surface. Generally the longer a water mass stays isolated from the surface and/or is mixed with different water masses, the less coupled changes on oxygen concentration and $O_{2,sat}$ will be. This is likely not a first order concern for the


NWES, being a highly dynamic system, characterised by short flushing times and intense mixing, both wind- and tidal-driven, that effectively resets surface oxygen towards equilibrium with atmospheric pO2 every winter, or on shorter timescales in permanently mixed regions. However, future studies addressing different regions must take this into account, e.g. by using different metrics such as True Oxygen Utilisation (TOU, Ito et al. 2004) or Estimated Oxygen Utilisation (EOU, Duteil et al. 2013).


When it comes to the impacts of deoxygenatios, $O_2$ partial pressure, rather than concentration, determines oxygen supply in aquatic organisms (Hofmann et al., 2011, Clarke et al. 2021). Warming, by increasing metabolic demand, does exacerbate the impacts of deoxygenation (Rubalcaba et al. 2020). Temperature and oxygen should be considered together when quantifying future habitat suitability and a number of metabolic indexes have been developed that describe this (e.g. Deutsch et al. 2020, Clarke et al. 2021). Here our focus is limited to deoxygenation, rather than its impacts. However, since


deoxygenation is concerning because of its potential impacts on organisms, it is important that future studies take this into account.

## 5 Conclusions

At present few downscaled climate projections exist for the NWES, and even fewer that include biogeochemistry. By
producing two additional climate model runs we expanded on previous results on circulation (Holt et al. 2018) and near-bed oxygen (Wakelin et al. 2020) with the aim of improving the understanding of near-bed oxygen fate and controls in this region.

All of our ensemble members predict oxygen decline throughout NWES but also mitigating effects, due to increased primary production, in shallow coastal regions. Under sustained enough warming the eastern part of the North Sea and the
Norwegian Trench / Skagerrak complex are more vulnerable to oxygen depletion than elsewhere on the shelf; this is tied to increased stratification, fostered by a circulation change that limits ocean-shelf exchange.

This work serves not only to improve on the current understanding of the fate and controls of near-bed oxygen in the NWES, but also to stress how, especially in highly dynamic coastal and shelf environments, oxygen change can exhibit high spatial heterogeneity, much more than would be expected by the effects of warming on solubility alone. Several studies highlighted
how downscaling can improve the representation of coastal and shelf processes that are not adequately resolved in coarse global models. The drawback is that, more often than not, too few regional downscaled simulations are available to adequately characterise all sources of variability and quantify uncertainty, as is possible with global models. This is the case for this study also where we used a limited number of realizations (three) of the same coupled model suite and only one climate change scenario, and albeit we did explore variability to some degree, this can by no means be considered a robust
assessment of expected change. We suggest that future efforts should be directed at collating ensembles of regional climate model projections (similarly to what done within the Climate Model Intercomparison Project, CMIP), including the biogeochemistry component, for the purpose of studying marine climate and ecosystem impacts and controls, including those on oxygen. Such an effort should adequately sample scenario, internal and model variability. Our observation that differences in downscaling procedure, model forcings and parameterisation can significantly affect projected trends and the
representation of biogeochemical cycles highlights the importance of this.

**Code availability.** NEMO and ERSEM are both free and open source, their code can be retrieved from the PML github repository (github.com/pmlmodelling). NEMO was used in a configuration called AMM7 that models the NWES domain.

**Data Availability.** The physics model data for GFDL and IPSL are available here: https://gws-access.jasmin.ac.uk/public/recicle/, the physics model data for HADGEM are available here: https://zenodo.org/record/3953801#.ZeusdNLMJyZ. For the biogeochemical model data for all three ensemble members, the data behind the plots of the manuscript are available here https://zenodo.org/records/8283355, more data are available from

the corresponding author upon request. The rest of the data used in this study were published by their authors as cited in the paper.

**Author Contributions.** GG performed the analyses and wrote the main body of text, GG, YA and SW designed the analysis framework, all authors were involved in revising the manuscript, all authors contributed to model development and ran the simulations.

**Financial support.** This project has received funding from the European Union's Horizon 2020 research and innovation programme under grant agreement No 820989 (project COMFORT, Our common future ocean in the Earth system – quantifying coupled cycles of carbon, oxygen, and nutrients for determining and achieving safe operating spaces with respect to tipping points), and from the NERC projects RECICLE (Resolving climate impacts on shelf and coastal sea ecosystems NE/M004120/1 and NE/M003477/2), FOCUS (Future states of the global coastal ocean: understanding for solutions – NE/X006271/1) and CLASS (Climate Linked Atlantic Sector Science – NE/R015953/1). This work used the ARCHER and ARCHER2 UK National Supercomputing Service (http://www.archer2.ac.uk, last access: 26 April 2023).

**Competing interests.** The authors declare that they have no conflict of interest.

**Disclaimer.** The work reflects only the author's/authors' view; the European Commission and their executive agency are not responsible for any use that may be made of the information the work contains.

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
