# Peer review of "Multi-model comparison of trends and controls of near-bed oxygen concentration on the Northwest European Continental Shelf under climate change"

_EGUsphere, 2023_

## Referee Comment (RC2)

Giovanni Galli et al. have evaluated the trends and controls of $O_2$ changes due to biogeochemical and physical changes in the NWES using model data for the 21st century. Unfortunately, the methods are unclear to me (ensemble description, analysis of $O_2$ change) as well as the research questions/novelty of the study. I agree it is important to assess biogeochemical changes and drivers (and their uncertainty) in such a heavily exploited and strongly changing region. As I could not follow the methods everywhere, I can only give an incomplete review of the manuscript at this stage. Here are some comments that may help to improve the manuscript:

**Comments**
-   The title does not really seem to cover the results (How about ' 21st century trends and controls of near-bed oxygen change on the Northwest European Continental Shelf' or so?
-   Abstract: Could you quantify some of your statements? What is new here?
-   l85-92: You make an elaborate comparison here, but then also underlines the limited usability of Kwiatkowski et al. (2020). I would just highlight the limitation of ESMs to quantify this region if you like, but not compare.
-   L116-122: which reference(s) is this all based on?
-   L128: for regional models boundary conditions are also highly relevant. Spinup-times could also be mentioned here. Do you wish to provide an exhaustive list here?
-   L131: Why not CMIP6?
-   L 134: you did not investigate ecosystem reponses?
-   L130-132: I get the feeling here you used three models and then ran 3 ensembles within each model (namely by using slightly different CMIP5 forcings), can you clarify this already here?
-   Introduction: your final paragraph (lines 126-136) describes what your new contribution is. However, it is unclear at this stage what extra model variability you argue to have covered (and important to note that there are several sources of uncertainty, scenario, model variability, model uncertainty, see for example Fig. 3 in https://agupubs.onlinelibrary.wiley.com/doi/10.1002/2015GB005338). Also, your research questions are not so clear to me. Why did you focus on the near-bed $O_2$ specifically, why not the whole water column and then near-bed as a separate focus?
-   Line 140: is a 10-year spin-up enough? In ESMs a few hundred years is more common. What drift do you have in your variables during this spinup? If significant, drift should be subtracted from the data at the least (and a thorough discussion should be provided why you can still use the data).
-   Sect. 2.1: I do not follow. There are 3 models which all are part of the NEMO-ERSEM model suite (so 3 times almost the same model?). Are these the 3 members then? Which you then forced with ESM data (from GFDL, IPSL and HADGEM)? What are the parent ESMs then (as its says that the boundary conditions of these 3 ESMS are taken from parent ESMs (line 152), these CMIP5 data are fully coupled ESMs without boundary conditions except towards space)? So, do you have 9 model runs in total (3*3)? When you write about 'all models' in line 155 you seem to be discussing the ESMs as if you have been running the ESMs, but you used this NEMO-ERSEM setup, right? Anyway, I do not follow. Alternating between the word member and model might be inconsistently done? Maybe a table? What happens in the forcing in the 21st century (e.g., wind/freshwater forcing changes?)

- L 150 and what are the ECS then of these models?
- Sect. 2.3: SS_t0 is not defined here? How is this approach different from AOU (Apparent Oxygen Utilization) or even better TOU (True Oxygen Utilization)? You open with that $O_2$ change have 3 different components, but then you can only separate into 2, right? Namely the temperature effect through its effect on $O_2$ saturation and then biology+circulation as the 'other' term (which is like in AOU and I am not aware of a method that can distuingish all 3). Calculating $O_2^{sat}$ and the contribution of $O_2$ from circulation+bio is a simple calculation I would say, and I think the analysis should go beyond this and the correlations.
- Sect. 3.1: Why don't you bias-correct and only use the model/ensemble trends (like you actually do in e.g. Fig. 3), considering the significant biases? Then the absolute errors are less important and can go into an appendix or so. You seem to have done so anyway for (part of?) your analysis (mentioned in line 276 and caption Fig. 4 only...).
- Fig. 2: based on the text units here are standard deviations? Maybe just use the full names instead of nurmsd and nbias? Or just call them Root mean square and bias and say that they are normalized? I think it would be good to get the equations from Jolliff et al. (2009) or to use more commonly used metrics like RSS?
- L 259: here you for the first time use the word downscaling, this should be introduced in the methods section.
- Fig. 3: If you would plot instead of a change over time a change at a certain global warming level (countering the differences in ECS, see Hausfather et al. (2022); 10.1038/d41586-022-01192-2), your model differences will likely be smaller? Would that be a more meaningful way of assessing model differences as showing differences in warming is inherent to choosing models with different ECS?
- Can Sect. 3.4-6 be merged?
- Sect. 3: I was actually a bit surprised about the section titles here, and it would be good to introduce the reader earlier what you will exactly cover in your results section to answer your research questions.
- Sect. 4: this mostly sounds like a conclusions/summarizing section except for the last paragraph. Please try to discuss limitations of your methods, implications, compare to other studies that may show something else? You find many confirmations/consistencies which is fine but makes your work sound less novel or complementary. What other stressors does the near-bed ecosystem experience (trawling/pollution?)?
- L 430: how does your study highlight this? Could you show your regional/downscaled model runs are superior to the ESM output? Same in line 441.
- You mention that you asses 'ecosystem impacts' throughout the manuscript, but I would say you mostly assessed a range of physical and biogeochemical changes and the possible drivers of the $O_2$ changes.
- L450-451: reference?
- Sect. 5: I do not see so well how this section connects to your results. Please quantify your results and focus your conclusions on the answers to your research question(s). E.g., your conclusions and abstract text are quite different while one would expect them to cover very similar statements. Sect. 3.9 is not discussed or concluded upon.
- correlation is not causation

**Minor remarks:**
- Some spelling errors that can be captured by any spellchecker are still in the text
- L76: possibly? Sometimes? Regularly?
- L185: limit validation?

---

## Author Comment (AC1)

**REVIEWER 1**

GENERAL COMMENTS:

REV1:

The manuscript investigates the processes driving near-bed oxygen changes on the Northwest European Continental Shelf under a high-emissions climate change scenario, with a focus on the intermodel uncertainties in these processes and their effects on oxygen. This work extends and qualifies the results of a previous study (Wakelin et al., 2020) by adding two additional sets of regionally downscaled model projections within the high-emissions forcing scenario (RCP 8.5).

Ocean deoxygenation and coastal hypoxia under climate change pose a serious threat to marine ecosystems. Robust understanding and projection of these processes is important for effective adaptation of ecosystem services. Given the lack of skill of coarse resolution global ESMs in coastal regions, regional downscaling of ESM projections will likely play a critical role in exploring this topic.

Although these additional model simulations provide valuable new insights into the fate of the oxygen in the region, some of the main conclusions reached by the authors are not well supported by the evidence presented. The scope of the study is not well defined and the manuscript overall lacks focus and rigor. While the scientific premise of the study is valuable, major revisions are required for this work to be fit for publication in Biogeosciences.

ANSWER:

We thank Reviewer 1 for the useful comments. We thoroughly revised the manuscript according to all reviewers' comments, we took special care in providing additional support to our conclusions, we better defined the scope of the study and its limitations.

SPECIFIC COMMENTS:

REV1:

(1)

A major result of the paper is the attribution of the deoxygenation hotspot in the Norwegian Trench to a relaxation reversal of the Norwegian Trench Current; but this interpretation is not well supported or well argued. The authors argue that (1) a relaxation of the advective current causes a freshening of the shelf region causing increased stratification, and (2) correlation suggests that the increased stratification is responsible for deoxygenation. Holt et al (2018) argue that changes in stratification are responsible for the relaxation of the current, opposite to the authors' explanation. In most cases, an increase in stratification would come from surface warming and precipitation changes; this null hypothesis should be disproved before seeking alternative explanations.

(2)

It is also not clear in the results whether vertical mixing or horizontal advective transport is dominating oxygen supply to the Norwegian Trench region, which should guide the conclusions made. Note that Wakelin et al (2020) do link reduced current to a recoupling of export with nearbed respiration; perhaps this is connected to the change in sign of correlation between SS and stratification (320).

(3)

Lastly, 'tight coupling' in Figure 11 is not necessarily convincing by eye. A stronger link has to be made.

ANSWER:

(1)

We appreciate how this may have not been entirely clear in the text but our conclusions about the causes of the relaxation of the WNT current are not at odds with Holt et al. 2018. In both works it is increased stratification at the northern entrance of the trench that reduces oceanic inflow into the North Sea, this in turn increases retention of fresh water from continental Europe and the Baltic within the North Sea, driving freshening (Holt et al. fig1e,f, this study, fig 3.) and a further increase in stratification in the North Sea. Then there certainly is a component of the increase in stratification due to the atmospheric temperature forcing, but this cannot explain the hotspot of increased stratification as surface warming is homogeneous across the domain (fig. 3.).

We revised the manuscript to make all of this clearer.

(2)

We acknowledge the reviewer's comment, however in our model configuration, the lateral transport of oxygen from the Baltic open boundary does not change in time. We acknowledge that this was not clarified in the manuscript, but in GFDL and IPSL, the Baltic open boundary has fixed climatological values for all tracers, including oxygen. This choice was made because ESMs are scarcely reliable for an enclosed sea such as the Baltic. This ensures that the deoxygenation signal we detected in the Norwegian Trench does not originate from the Baltic boundary through lateral transport.

HADGEM also uses a fixed climatology at the boundary for both biogeochemical variables (including oxygen) and freshwater input, with the difference that the Baltic boundary is treated like a river, rather than an open boundary.

Both boundary treatment choices do have some limitations, however they also rule out lateral transport from the Baltic as the source of the deoxygenation signal in the Norwegian Trench.

We clarified this in the methods, and added some discussion about the limitation from having a climatological boundary at the Baltic while being able to rule out lateral transport as a contributing factor for deoxygenation hotspots.

(3)

We complemented section 3.9 (now 3.7) with correlation coefficients for the analysed timeseries and revised the text according to the results.

REV1:

The title of the manuscript suggests that the focus of the paper is on 'intra-scenario variability'; however, it is unclear what the scope of this is and how effectively it can actually be investigated with available tools. Uncertainty in ESM climate projections (and by extension, downscaled projections) fall broadly into three categories, regarding (1) internal model variability, (2) intermodel uncertainty, (3) and scenario uncertainty. The term 'intra-scenario' would suggest that

you look at both internal variability and intermodel uncertainty, which is not really the case. Due to the small sample size (three models) and inconsistencies in the model and methods used for downscaling in the older HADGEM run versus the IPSL and GFDL simulations, neither internal variability nor intermodel uncertainty is well sampled nor well isolated. Perhaps the term 'multi-model comparison' used in the abstract is more appropriate here. This is already addressed somewhat in the introduction (125-135), but should be clarified and given more thought. Claims like "we added an intra-scenario variability dimension (375)" are unclear and misleading, and should be changed.

ANSWER:

We appreciate the focus on "Intra-scenario variability" may be misleading, and we concur with the reviewer that the scope here is to compare the projections of oxygen from the small "multi-model" ensemble. Therefore, we changed "Intra-scenario variability" in the title and throughout the text with "multi-model comparison", or deleted it, revised the introduction by mentioning the categories of uncertainty in projections and by more clearly stating the aim of the study, shifting the focus away from variability estimation. We revised the discussion by stating which sources of variability were not addressed in this study. We revised the conclusions highlighting the importance of sampling different sources of variability while building regional climate model ensembles.

REV1:

Throughout the study, the authors claim that oxygen changes in the study region across the three simulations scale with the climate sensitivity of the parent ESMs. If quantifications of these sensitivities are available, they should be presented here. Additionally, an issue with this claim is that the differences in downscaling methods for HADGEM vs the IPSL and GFDL simulations provide uncontrolled degrees of freedom. The authors should provide an argument whether the differences in downscaling techniques should significantly impact the magnitude of oxygen changes. If possible, the authors could run some short sensitivity experiments using the new (used for IPSL, GFDL) setup to test sensitivity to e.g. vertical resolution.

ANSWER:

We added the estimates for the global equilibrium climate sensitivity of the three parent ESMs (these are 4.59, 4.12 and 2.39K for HADGEM2-ES, IPSL-CM5A-MR and GFDL-ESM2G respectively).

We also added in section 4 a more detailed discussion on how the different downscaling methods from HADGEM may influence the results. Unfortunately producing conclusive evidence requires ad-hoc experiments, and as the reviewer suggest, this can be quite an expensive task that is not always feasible for multiple reasons, including availability of resources. While we agree that the differences in the model set-up may play a role in the dynamic, these will not be the driving cause of the patterns projected by the model.

Nonetheless, despite some noticeable different responses, the bulk of the behaviour of our ensemble members is still coherent with the tested climate change intensities. This we think shows that our results are still robust with respect to the [limited] model variability represented in our ensemble.

REV1:

In the model used by Wakelin et al (2020), oxygen is not included in open boundary conditions of the regional model so that changes in open ocean oxygen is not included. Is this the case here? This is very important for how the results may be interpreted and should be documented carefully.

ANSWER:

We appreciate this was not explicitly mentioned in the text. In our IPSL and GFDL members oxygen is indeed included in the open ocean and Baltic boundaries, whilst Wakelin et al. (2020), that is our HADGEM, uses a zero gradient-scheme (i.e. boundary concentration equals concentration inside the domain) for most biogeochemical tracers, including oxygen, at the open ocean boundary. The only tracers that are forced with external data at the open ocean boundary are nutrients and inorganic carbon. At the Baltic boundary HADGEM uses climatological values for all tracers, including oxygen.

We improved the Ensemble description in the manuscript, so that all boundary schemes are clearly described.

The impact of the different treatment of the boundary will largely impact the open ocean part of the domain (that is excluded from the analysis), while the shallow depth and intense winter mixing of the NWES makes so that ocean-atmosphere exchange will reset the oxygen to saturation every winter, or more frequently, throughout the water column, so that oxygen on shelf is scarcely coupled with oceanic oxygen.

REV1:

The authors need to be careful when interpreting correlation as causation. Correlations are only meaningful when there is a process that can explain the relationship. Please be thorough about when a physical/ biogeochemical mechanism can explain a correlation and when a correlation cannot be explained. For example, why would you have a positive correlation between SS and stratification in some regions (Fig. 7)? If strong but erroneous correlations are prevalent, why can we still trust the results? The authors should also provide a discussion of any covariances that may influence the results (e.g. between temperature, stratification, respiration, NPP)

ANSWER:

We agree with the reviewer that correlation does not automatically imply causation, and we can support our interpretation by improving the presentation of the results and the discussion of the attribution of correlations. In particular we added detail about:

[1] corr(SSO2, Tatm)>0 in southern coastal regions, all members, covariance explained by increasing NPP,

[2] corr(SSO2, PEA)>0 in coastal regions, covariance mediated by the seasonality in NPP.

[3] corr(SSO2, Tatm)<0 in the Trench and Eastern North Sea, all members, (new results without detrending see later, covariation with increasing PEA),

[4] corr(BResp, SSO2)>0 in the Norwegian Trench, IPSL, covariance explained by decreasing BResp, due to decreasing NPP, together with decreasing SSO2 due to increased stratification (no strong direct causal link).

REV1:

In calculating correlations, the long-term trend is removed. I see how this avoids false positives, but how can you assess the drivers of forced changes after removing the long term trends? In this case, it seems that correlations just classify the drivers of short-term variability, which is not what you purport to be investigating. Please explain/ clarify.

ANSWER:

We appreciate this may be of concern regarding our methodology. While analysing the data we did conduct exploratory analyses where trends were not removed, which only resulted in slight improvements of some detected correlations, with no relevant changes in sign. We concluded that trend removal was in this case the most conservative practice.

This perhaps could be justified in systems with short turnover rates where drivers of short- and long-term trend overlap. For example, warming reduces oxygen solubility both on the long-term, through increasing mean temperatures, and on short-term, e.g. during summer months. Or increasing NPP produces oxygen both on the short-term, during a bloom, and the long-term (if coupled with enough mixing) if the productivity of a region increases over time.

Nonetheless, we see a solid point can indeed be made in favour of retaining the trend when calculating correlations, if the aim is explaining the trend, and taking care that, when interpreting results, some patterns will be explained by covariances rather than causal links (false positives). This is what we did. As for the revised results the only relevant changes are:

1) Negative correlation between O2sat and atmospheric temperature in the Norwegian Trench (all members, instead of non-significant).

2) Negative correlation between SSO2 and atmospheric temperature in the Norwegian Trench and eastern part of the North Sea (HADGEM and IPSL, instead of non-significant).

3) Negative correlation between SSO2 and PEA along the Norwegian Trench (HADGEM and IPSL, instead of non-significant)

for 1) and 3) a case can be made for a causal link, for 2) the pattern is more easily explained by covariance with PEA.

None of these change our conclusions substantially but 3) removes the need for Fig 8. "running correlations between SSO2 and PEA mediated over the Norwegian Trench".

The correlations involving biogeochemical variables (NPP, BResp) didn't show any significant change.

We attach a revised version of the part of results that changed for a more complete exposition.

REV1:

What is gained by decomposition in section 2.3 as opposed to a traditional O2sat, AOU decomposition? Why is O2phys-ch (O2sat scaled by the initial saturation state) a more meaningful metric than O2sat? The authors end up using O2sat and SS (a.k.a 1-AOU) anyway, so this section can be removed entirely.

ANSWER:

Please note that we re-worked the methods section according to Reviewer3's comments, as a result the definition of ΔO2_other changed slightly and O2,phy-ch,t is no longer present, the comment still applies though.

The ΔO2_phys-ch and ΔO2_other metrics we presented in the methods are indeed related to the traditional O2sat, AOU decomposition, with the difference that they describe the partitioning of oxygen change relative to a reference period, rather than the distance from equilibrium at any specific moment.

This renders them interesting as metrics because they are directly comparable, being both $\Delta$ concentrations (unlike Osat (concentration), AOU ($\Delta$)) and they sum up to the total $\Delta O2$. This allows to quantify how much of the observed change can be attributed to each component.

AOU estimates oxygen consumption (and production) since a water parcel was last in contact with the atmosphere, assuming Osat doesn't change. Our metrics, by explicitly considering changes in Osat, allow to partition oxygen change into the two separate components.

We included a section in the methods explaining this, the relation between our metrics and AOU, and the hypotheses and limitation of both methods.

Results about $\Delta O2$,phys-ch and $\Delta O2$,other are presented in section 3.4 Contributions to near-bed oxygen change.

REV1:

How are there negative values in the root-mean-square distance calculation (Fig. 2)? Need to provide formulae here for nbias and nurmsd.

ANSWER:

we appreciate the metrics from Jollif et al. may not be as widely known as others, we addressed this by adding their definition in the methods (although we merely described the equations, rather than writing them down, as they are indeed trivial). As for the negative values of nurmsd, they arise by multiplying rmsd by the sign of the difference of model and data stds, so that a negative value indicates that the model's std is lower than that of the observations, and vice-versa for positive values. We explained also this in the text.

REV1:

Bias-correction for hypoxia measurements should be included in methods

ANSWER:

we included bias correction procedure in the methods.

TECHNICAL COMMENTS:

REV1:

Grammatical errors and inconsistent capitalization throughout. Please proofread carefully.

ANSWER:

We carefully proof-read the manuscript and corrected errors and capitalization.

REV1:

In all figures, panel labels need to be included.

ANSWER:

Panel labels have been added to all figures.

REV1:

Use consistent terminology for region names. Is the Danish strait the same as Skagerrak? Eastern North Sea is referenced throughout but not delineated on the map in Fig 1.

ANSWER:

We replaced Danish strait with Skagerrak, we also replaced 'Eastern North Sea' with 'eastern part of the North Sea', or similar, throughout the text.

REV1:

Nearly all instances of 'in fact' can be removed

ANSWER:

all instances of 'in fact have been removed or replaced'

**3.5 Physical controls of oxygen change: temperature and stratification**

Changes in $\Delta O_{2,phy\text{-}ch}$ and $O_{2,sat}$ are, for the greatest part, explained by warming (correlation between $O_{2,sat}$ and near-bed T ~-1 everywhere in all models, not shown). The driver of this is the temperature atmospheric forcing (Fig. 6) that in all models displays strong negative correlation with near-bed $O_{2,sat}$.

Conversely atmospheric temperature correlates positively with $SS_{O2}$ in coastal regions around the British Isles and continental Europe (including the Southern North Sea Channel and Irish Sea) in all models. This appears to be mediated by a covariation with increasing NPP in these well mixed areas fuelling oxygen production (see section 3.6). Positive correlation between $SS_{O2}$ and atmospheric temperature in the Central and Northern North Sea, which stratify seasonally, in IPSL and GFDL may instead be mediated by covariation with decreasing respiration in these areas, which is due to decreasing NPP (see section 3.6).

Atmospheric temperature and $SS_{O2}$ instead are negatively correlated in IPSL and HADGEM in the regions of the deoxygenation hotspots, Norwegian Trench and eastern part of the North Sea. This is mediated, for both models, by covariation with increasing stratification in these regions (see below).

[Figure]

**Fig. 6.** correlation between Temperature atmospheric forcing and near-bed O2,sat and $SS_{O2}$.

The North Sea hotspots of oxygen decline in HADGEM and IPSL coincide with enhanced stratification hotspots and indeed $SS_{O2}$ and potential energy anomaly (PEA - an indicator of stratification de Boer et al. 2008) are, in both ensemble members, strongly negatively correlated in this area (Fig. 7);

GFDL on the other hand only shows a moderate increase in stratification and no significant hotspots, with a weaker correlation between $SS_{O2}$ and PEA than in the other two models. The main driver of stratification along the Norwegian Trench and in the eastern part of the North Sea is, for

all models, surface salinity, that is strongly negatively correlated with PEA there and over much of the domain, especially in HADGEM and IPSL.

The positive correlation between PEA and $SS_{O2}$ in coastal areas in the southern North Sea and around the British Isles (observed in all ensemble members) appears to be mediated by the seasonality of primary productivity. These shallow regions experience strong tides and remain well mixed year-round (PEA barely changes in the long term). Here stratification is not a meaningful indicator of vertical oxygen transport. However. The highest PEA values do happen in the summer months, when also NPP peaks, producing oxygen that contributes to high $SS_{O2}$ values, while the opposite is true during winter; hence the positive correlation.

[Figure]

**Fig. 7.** Change in potential energy anomaly (PEA) and correlation between PEA and surface salinity and PEA and $SS_{O2}$.

---

## Author Comment (AC2)

**REVIEWER 2**

GENERAL COMMENTS

REV2:

Giovanni Galli et al. have evaluated the trends and controls of O2 changes due to biogeochemical and physical changes in the NWES using model data for the 21st century. Unfortunately, the methods are unclear to me (ensemble description, analysis of O2 change) as well as the research questions/novelty of the study. I agree it is important to assess biogeochemical changes and drivers (and their uncertainty) in such a heavily exploited and strongly changing region. As I could not follow the methods everywhere, I can only give an incomplete review of the manuscript at this stage. Here are some comments that may help to improve the manuscript:

ANSWER:

We thank Reviewer 2 for the useful comments, we thoroughly revised the manuscript according to all reviewers' comments. Among other change we improved the ensemble description removing possible sources of ambiguity as well as the methods description and stated the study objectives more clearly.

SPECIFIC COMMENTS

REV2:

The title does not really seem to cover the results (How about ' 21st century trends and controls of near-bed oxygen change on the Northwest European Continental Shelf' or so?

ANSWER:

We acknowledge "intra-scenario variability" was misplaced here. We would refrain to use '21st century trends' because our ensemble is not large enough to produce a robust assessment of expected trends, or to quantify uncertainties. However, under Reviewer1's suggestion, we changed 'intra-scenario variability' to 'multi-model comparison' to reflect this concern.

REV2:

Abstract: Could you quantify some of your statements? What is new here?

ANSWER:

We added some quantification of results in the abstract, and some more in the results section.

We also improved the last paragraph of the abstract by stating the novelty and relevance of this study. In short, we expand on Wakelin et al (2020) showing that the projections of near-bed oxygen presented there are robust in a multi model context and that trends and drivers of change remain coherent at different warming intensities. Finally we want to assess the impact of the change in circulation presented by Holt et al. (2018) on oxygen change similarly across models. These are new findings that allow a better understanding of the projection of near-bed oxygen in the NWES and its drivers. We believe (and reviewer 1 seem to concur) that these are important questions to

assess the impact of climate change on shelf seas and plan the needed adaptation, and it may suggest directions for future research.

REV2:

l85-92: You make an elaborate comparison here, but then also underlines the limited usability of Kwiatkowski et al. (2020). I would just highlight the limitation of ESMs to quantify this region if you like, but not compare.

ANSWER:

Our aim here was not really comparing the two results, which we believe may be equally valid, but showing that when looking at oxygen change results may vary also according to the spatial scales and domains being analysed (coastal vs shelf). Then global ESMs have indeed some known limitations in resolving coastal and shelf processes and we rephrased the part of text when we explain this to make it clearer and more specific.

REV2:

L116-122: which reference(s) is this all based on?

ANSWER:

It's all Wakelin et al. 2020, we are summarising the results of that paper and that took some text, we appreciate this may not have been entirely clear, so we repeated the citation in the text.

REV2:

L128: for regional models boundary conditions are also highly relevant. Spinup-times could also be mentioned here. Do you wish to provide an exhaustive list here?

ANSWER:

We rephrased the sentence so that we refer to the components of variability (Frölicher et al. 2016): internal, model and scenario variability, and provided some examples of variability sources for internal and model variability. Since this particular sentence is general, it applies to both regional and global models, to atmosphere, ocean or land, it may not be advisable to mention all possible sources of variability.

REV2:

L131: Why not CMIP6?

ANSWER:

These simulations were implemented before outputs from CMIP6 were available. While we acknowledge the differences between the CMIP5 and the CMIP6 scenarios, CMIP5 still provide a useful set of climate scenarios that are still valuable and that can be associated to the CMIP6 ones.

REV2:

L 134: you did not investigate ecosystem responses?

ANSWER:

We rephrased the sentence, to make clear that we look at near-bed oxygen change.

REV2:

L130-132: I get the feeling here you used three models and then ran 3 ensembles within each model (namely by using slightly different CMIP5 forcings), can you clarify this already here?

ANSWER:

We have 3 downscaled ensemble members, each one is forced with one single CMIP5 ESM. We appreciate this may not have been clear in the text so we revised it accordingly.

REV2:

Introduction: your final paragraph (lines 126-136) describes what your new contribution is. However, it is unclear at this stage what extra model variability you argue to have covered (and important to note that there are several sources of uncertainty, scenario, model variability, model uncertainty, see for example Fig. 3 in hdps://agupubs.onlinelibrary.wiley.com/doi/10.1002/2015Gti005338). Also, your research questions are not so clear to me. Why did you focus on the near-bed O2 specifically, why not the whole water column and then near-bed as a separate focus?

ANSWER:

We acknowledge the claim of addressing [comprehensively] some aspect of variability may be misplaced here. We improved the final paragraphs of the introduction where we describe the aim of the study so that we don't refer to this. We also changed in the title and throughout the text the term "intra-scenario variability" to "multi-model comparison", which is perhaps more appropriate here. We also stated in the discussion which sources of variability our study fails to address.

We also improved the exposition of aim of the study that is further qualifying near-bed oxygen projections, their trends, and drivers in the NWES in a multi-model context, and assessing the impact of circulation changes on near-bed oxygen similarly in a multi-model context. All of which represent understudied questions.

We choose to focus on near-bed oxygen because the bottom layer is more vulnerable to deoxygenation than the surface, due to the seasonal decoupling from the atmosphere. Furthermore, near-bed oxygen dictates habitat suitability for benthic sessile or scarcely motile species and it is used as an indicator of eutrophication in the North Sea (Devlin et al. 2023).

We also added to the introduction some more explanation on why the focus on near-bed oxygen.

REV2:

Line 140: is a 10-year spin-up enough? In ESMs a few hundred years is more common. What drift do you have in your variables during this spinup? If significant, drif should be subtracted from the data at the least (and a thorough discussion should be provided why you can still use the data).

ANSWER:

While a 10y spinup is not enough for Earth System Models, it is for the regional model used here and it is in fact common practice (e.g. Tinker et al. 2014, Holt et al. 2018, Ciavatta et al. 2018). The North Sea has a flushing time of 2-4 years, the Norwegian Trench about 100d (Blaas et al. 2001), depths are relatively shallow and seasonal mixing is intense throughout the water column in most locations; all of this concurs in making these relatively short spinup times acceptable for this domain. Indeed we did not observe appreciable drifts in the model during the spinup period. In addition to that the biogeochemical initial conditions, which include the slowest components of the system (i.e. the benthos) are initialised with the reanalysis from Ciavatta et al. 2018, which ran for an additional 10y, including its own spin-up. We added a couple of lines explaining this.

REV2:

Sect. 2.1: I do not follow. There are 3 models which all are part of the NEMO-ERSEM model suite (so 3 times almost the same model?). Are these the 3 members then? Which you then forced with ESM data (from GFDL, IPSL and HADGEM)? What are the parent ESMs then (as its says that the boundary conditions of these 3 ESMS are taken from parent ESMs (line 152), these CMIP5 data are fully coupled ESMs without boundary conditions except towards space)? So, do you have 9 model runs in total (3*3)? When you write about 'all models' in line 155 you seem to be discussing the ESMs as if you have been running the ESMs, but you used this NEMO-ERSEM setup, right? Anyway, I do not follow. Alternating between the word member and model might be inconsistently done? Maybe a table? What happens in the forcing in the 21st century (e.g., wind/freshwater forcing changes?)

ANSWER:

We have 3 downscaled ensemble members that all use the NEMO-ERSEM suite, each one is forced with boundary conditions from one of three fully coupled CMIP5 ESMs, the "parent ESMs", which are GFDL-ESM2G, IPSL-CM5A-MR and HADGEM2-ES. So we have 3 downscaled ensemble members in total, which, for brevity, we call GFDL, IPSL and HADGEM (instead of "the downscaled ensemble member forced with boundary conditions from GFDL-ESM2G, IPSL-CM5A-MR or HADGEM2-ES"). We appreciate that our presentation might have been confusing, we revised it in order to make it clearer, and we took care of being consistent in the use of ensemble member instead of model.

As for the forcings in the 21st century they are as described in the text: those available from the CMIP5 (e.g. wind, lateral boundary conditions, etc.) are from CMIP5 (both historical, up to 2005, and climate runs), those not available from CMIP5 are from other sources (e.g. river nutrient loads are from a reanalysis, Ciavatta et al. (2018), multiplied by river discharge). We also revised the Ensemble description in the text in order to provide a clearer and more comprehensive list of all the forcing fields.

REV2:

L 150 and what are the ECS then of these models?

ANSWER:

We added to the text the estimates of the ECSs of the ESMs, thes are 4.59, 4.12 and 2.39K for HADGEM2-ES, IPSL-CM5A-MR and GFDL-ESM2G respectively (Andrews et al., 2012, Dufresne et al., 2013).

REV2:

Sect. 2.3: SS_t0 is not defined here? How is this approach different from AOU (Apparent Oxygen Utilization) or even better TOU (True Oxygen Utilization)? You open with that O2 change have 3 different components, but then you can only separate into 2, right? Namely the temperature effect through its effect on O2 saturation and then biology+circulation as the 'other' term (which is like in AOU and I am not aware of a method that can distuingish all 3). Calculating O2 sat and the contribution of O2 from circulation+bio is a simple calculation I would say, and I think the analysis should go beyond this and the correlations.

ANSWER:

Please note we reworked the methods section extensively, following Reviewer3's suggestions. The comment still applies though.

Our approach is indeed related to the classic AOU / O2sat decomposition, with the difference that our approach quantifies components of change at a point in space relative to a reference time period, whereas AOU measures the time-integrated amount of oxygen consumed since a water parcel has left the surface.

Whereas AOU implicitly assumes that O2sat of a water parcel doesn't change since its last contact with the atmosphere, our method explicitly accommodates changing O2sat (due to e.g. ocean warming). This way ΔO2 can be explicitly partitioned in two components, one related to changing O2sat and one related to changing SSO2.

ΔO2phy-ch and ΔO2other are directly comparable measures, being both components of the total ΔO2, as opposed to AOU (Δ concentration) and O2sat (concentration).

As for other metrics such as True oxygen utilisation (TOU Ito et al. 2004) and Evaluated Oxygen Utilisation (EOU, Duteil et al. 2013), the first must be evaluated explicitly at runtime to define the preformed oxygen, which unfortunately we haven't implemented, and both address some known issues with AOU when O2 concentration and solubility are decoupled, which may happen e.g. when undersaturated water is subducted or when a water mass changes temperature away from the surface, or in the presence of sea ice. This is not quite the case in the NWES that is relatively shallow and well mixed, with intense ocean-atmosphere exchange. Which makes the assumptions behind AOU (and our method as well) a fair approximation. Overall, we felt the use of different metrics wouldn't have brought much improvement to the results. We added some lines in the discussion addressing the limitation of our method.

While we agree our analysis method is fairly simple, we still believe that this is still informative, and indeed we successfully exploited it to diagnose oxygen dynamics and controls in our ensemble members.

REV2:

Sect. 3.1: Why don't you bias-correct and only use the model/ensemble trends (like you actually do in e.g. Fig. 3), considering the significant biases? Then the absolute errors are less important and can go into an appendix or so. You seem to have done so anyway for (part of?) your analysis (mentioned in line 276 and caption Fig. 4 only…).

ANSWER:

We don't bias correct extensively (with the exception of the hypoxia estimation in fig. 4) because we almost exclusively look at delta concentrations and correlations and these are not influenced by bias. Instead we use bias correction when calculating hypoxia because when fixed low oxygen

thresholds are considered, absolute values are relevant. We added some lines to the manuscript to make this clearer.

REV2:

Fig. 2: based on the text units here are standard deviations? Maybe just use the full names instead of nurmsd and nbias? Or just call them Root mean square and bias and say that they are normalized? I think it would be good to get the equations from Jolliff et al. (2009) or to use more commonly used metrics like RSS?

ANSWER:

We don't indeed use normalised rmsd but normalised unbiased rmsd, which is normalised rmsd multiplied by the sign of the difference of model and data's stds, so that it can also assume negative values, which indicate that the model's std is smaller than that of the observations, and vice-versa for positive values. We explained this in the text, we didn't add in the full equation because that is indeed quite trivial.

REV2:

L 259: here you for the first time use the word downscaling, this should be introduced in the methods section.

ANSWER:

We mentioned "downscaling" several times throughout the text.

REV2:

Fig. 3: If you would plot instead of a change over time a change at a certain global warming level (countering the differences in ECS, see Hausfather et al. (2022); 10.1038/d41586-022-01192-2), your model differences will likely be smaller? Would that be a more meaningful way of assessing model differences as showing differences in warming is inherent to choosing models with different ECS?

ANSWER:

We appreciate this could be an interesting angle to look at our projections, and we did run some additional analyses to look into it (see attached document, WNT_and_Warming_Level.pdf).

However, warming level, either global or regional (over the downscaled domain), doesn't seem to be relevant for the development of the change in circulation in the North Sea, whose effects on near-bed oxygen are an important focus of our manuscript. In particular, both IPSL and GFDL, which are exactly the same model with different forcings, show regional atmospheric warming in excess of 2K, but whilst IPSL develops the circulation change, GFDL doesn't.

In the manuscript we improved the statement of the aim of the study, shifting the focus away from the evaluation of model variability and differences. In short, we aim at testing the system's response (with a special focus on effects of circulation changes) at different climate change intensities, as this is at present poorly constrained for many biogeochemical variables (oxygen included) in the NWES. That is why we chose models covering a wide range of ECSs.

REV2:

Can Sect. 3.4-6 be merged?

ANSWER:

we merged 3.5-6 and also 3.7-8, the new titles are as follows:

3.5 Physical controls of oxygen change: temperature and stratification

3.6 Biogeochemical controls of oxygen change: primary production and respiration

REV2:

Sect. 3: I was actually a bit surprised about the section titles here, and it would be good to introduce the reader earlier what you will exactly cover in your results section to answer your research questions.

ANSWER:

We improved the methods section in order to introduce more detail about the analyses we performed and whose results are presented in section 3.

REV2:

Sect. 4: this mostly sounds like a conclusions/summarizing section except for the last paragraph. Please try to discuss limitations of your methods, implications, compare to other studies that may show something else? You find many confirmations/consistencies which is fine but makes your work sound less novel or complementary. What other stressors does the near-bed ecosystem experience (trawling/pollution?)?

ANSWER:

We thoroughly re-wrote the discussion focussing on the implication of our results and on the limitation of our methods, and on the evidence available from the literature. We also discussed the combined impacts with other stressors in the conclusions section.

REV2:

L 430: how does your study highlight this? Could you show your regional/downscaled model runs are superior to the ESM output? Same in line 441.

ANSWER:

As we don't indeed provide a direct comparison with the oxygen field in the parent ESMs, we see how these statements were problematic. Providing such comparison would be, we believe, certainly interesting but also out of the scope of this paper. We rephrased the two sentences referring to literature rather than this study.

REV2:

You mention that you asses 'ecosystem impacts' throughout the manuscript, but I would say you mostly assessed a range of physical and biogeochemical changes and the possible drivers of the O2 changes.

ANSWER:

We rephrased all instances of 'ecosystem impacts' or similar in the text to make it clear that our assessment is limited to near-bed oxygen change.

REV2:

L450-451: reference?

ANSWER:

we rephrased the sentence and added a reference (Devlin et al. 2023, https://oap.ospar.org/en/ospar-assessments/quality-status-reports/qsr-2023/indicator-assessments/seafloor-dissolved-oxygen).

REV2:

Sect. 5: I do not see so well how this section connects to your results. Please quantify your results and focus your conclusions on the answers to your research question(s). E.g., your conclusions and abstract text are quite different while one would expect them to cover very similar statements. Sect. 3.9 is not discussed or concluded upon.

ANSWER:

We re-wrote the conclusions sections focussing on the answers to our research questions, we explicitly linked to what is stated in the abstract, and we improved the final recommendations. We also improved our discussion on how circulation change in the North Sea affects the development of deoxygenation hotspots.

REV2:

correlation is not causation

ANSWER:

We improved our results section by discussing in more detail the mechanisms that can observed correlations, including some that we failed to discuss earlier. This includes the discussion of covariances that determine correlations between variables also in the absence of a direct causal link. Here some examples:

[1] corr(SSO2, Tatm)>0 in southern coastal regions, all members, covariance explained by increasing NPP,

[2] corr(SSO2, PEA)>0 in coastal regions, covariance mediated by the seasonality in NPP.

[3] corr(SSO2, Tatm)<0 in the Trench and Eastern North Sea, all members, (new results without detrending under reviewer1's suggestion, covariation with increasing PEA),

[4] corr(BResp, SSO2)>0 in the Norwegian Trench, IPSL, covariance explained by decreasing BResp, due to decreasing NPP, together with decreasing SSO2 due to increased stratification (no strong direct causal link).

MINOR REMARKS

REV2:

Some spelling errors that can be captured by any spellchecker are still in the text

ANSWER:

We thoroughly revised the manuscript and corrected typos.

REV2:

L76: possibly? Sometimes? Regularly?

ANSWER:

frequently

REV2:

L185: limit validation?

ANSWER:

limit validation!

**Western Norwegian Trench current flux and warming levels.**

We compared (Fig. 1) the time evolution of the Western Norwegian Trench current flux (WNT) and warming levels in an ensemble of three downscaled ocean climate projections. The three members of the ensemble are forced with BDYs from one of three rcp8.5 CMIP5 ESMs: HadGEM2-ES, IPSL-CM5A-MR and GFDL-ESM2G. BDYs extracted from the ESMs and used for downscaling include atmospheric surface temperature. The three downscaled ensemble members are termed (HADGEM, IPSL and GFDL for brevity).

WNT is calculated as per Holt et al. (2018) and smoothed with a gaussian filter. Global Warming Levels (GWL) are calculated as the year at which the average atmospheric surface temperature in the three ESMs reaches the thresholds of +1, +2.5, +2.0, +2.5, …, +5°C relative to pre-industrial average. ESM global mean temperature time-series are smoothed with a 21y running mean prior to determination of the warming levels.
The Regional Warming Level (RWL) is calculated as the average atmospheric surface temperature change over the downscaled domain, smoothed with a 21y running mean, relative to the period 1990-2011.
Note that, since the reference periods change, RWL starts from 0, while GWL starts from 1.

A change-point in WNT is detected in both HADGEM and IPSL, but not in GFDL, starting approximately in the late 2020s, with a progressive weakening and reversal of the current flux. This happens at approximately +1.5-2.0°C GWL and +0.5-1.0°C RWL in both IPSL and HADGEM. However in GFDL no change in WNT is detected despite GWL up to +3.0°C and RWL up to +2.0°C.

We conclude that warming level, either regional or global is not a determining factor explaining the onset of the circulation change.

The circulation change is triggered by an increase in oceanic stratification at the northern entrance of the Norwegian Trench (Holt et al. 2018). It may hence be triggered by other factors not necessarily linearly related to warming level, e.g. ice melting. The representation of such factors in the ESMs has likely a crucial role here.

[Figure]

*Figure 1 WNT current flux (red), Global Warming Level (ΔT global, blue) and Regional Warming Level (ΔT region, yellow).*

---

## Author Comment (AC3)

**REVIEWER 3**

MAJOR COMMENT

REV3:

The flawed general premise is that somehow in situ $[O_2]^{sat}$ controls in situ $[O_2]$, while other mechanisms drive the saturation state. However, in the ocean interior and particularly near the seabed, far away from the surface, changes in solubility alone (from changes in temperature or salinity) should have zero effect on in situ $[O_2]$, except in the case where the solubility is reduced below the in situ $[O_2]$. If a parcel of water with salinity S, temperature T, and oxygen concentration $[O_2]$ was artificially cooled down, its $O_2$ solubility would increase, its $O_2$ saturation state would decrease, but its $O_2$ content would remain unchanged. While $\Delta[O_2]^{sat}$ may correlate well with $\Delta[O_2]$ in the real ocean and in marine biogeochemistry models, there is no causation.

ANSWER:

We understand the concerns that Reviewer 3 raises and, while we agree Reviewer 3 is essentially correct, we also believe that, in the specific case of the NWES, that is the object of our study, the less rigorous approach adopted here and in Wakelin et al. (2020) still represents a good approximation that is useful to understand the processes involved. Clearly this will need additional clarification in the text. Our argument is as follows:

There is a main assumption under the method we used, that the water column equilibrates with atmospheric $O_2$ on timescales short enough (every winter in seasonally stratified regions, more frequently in regions that are well-mixed year-round) to establish a causal link between $[O_2]^{sat}$ and $[O_2]$. This is similar to what is assumed in other widely used metrics like AOU.

This clearly may be far from true in the ocean interior, close to sea-ice interface or in upwelling areas, when a water parcel has been separated from the atmosphere long enough to degrade such causal link.

However the North Western European Shelf has characteristics that do not preclude using the decomposition adopted here: it is a highly dynamic system, characterised by relatively shallow depths, short residence times (2-4y for the North Sea, 100d for the Norwegian Trench) and, crucially, intense mixing, both wind- and tidally-driven; many regions (Irish Sea, English Channel, Southern North Sea) are known to be well mixed year-round, and the regions that stratify do so only seasonally. This effectively resets $[O_2]$ towards equilibrium with atmospheric $pO_2$ every winter, i.e. towards $[O_2]^{sat}$ (and SS toward 1), hence the causal link.

See also Ito et al. 2004, where differences between simulated preformed $[O_2]$ and $[O_2]^{sat}$ are small over much of the global ocean up to depths of 500-1000m and away from the poles.

Our main (and only) conclusion that concerns causal links between $[O_2]^{sat}$ and $[O_2]$ is that there is a component of $[O_2]$ change that is solely warming driven and mediated by the effect of temperature on solubility. This is quite trivial as a result, and it is already well established (e.g. Kwiatkowski et al. 2020, section 3.3 and fig 3).

For these reasons, we believe that our approach is justified, despite some limitations that we will better highlight in the paper to respond to the concerns of reviewer 3.

In the manuscript we improved the methods section by explicitly stating the hypotheses behind our method (and other metrics such as AOU), its limitations and potential pitfalls, and the reason why the method is still valid in the case of the NWES. Then, in the discussion, we mentioned again the

method's limitations and cited some existing alternative metrics (TOU, Ito et al. 2004, EOU, Duteil et al. 2013) that overcome them.

MINOR COMMENTS

REV3:

The authors notation is sometimes hard to parse and confusing, particularly when triple subscripts are used. I would recommend using a different notation, which I hope helps clarifying the comments presented here. (Note that the authors' notation is already different from the preceding work by Wakelin et al. (2020).) I recommend using a simpler symbol, such as $f = [O2] / [O2]^{sat}$ for the saturation state. Below I use the "0" subscript for "at t0", i.e., the 1990–2019 average, and the "1" subscript means "at t", i.e., the 2070–2099 average, so that for any quantity X, its 21st-century change is denoted by $\Delta X = X1 - X0$.

ANSWER:

While we recognise that other possible notations are in current use, the notation we use is at least partially consistent with published literature (e.g. Kwiatkowski et al. 2020, "$\Delta O_{2sat}$", Ito et al. 2004, "$O_{2,sat}$", Duteil et al. 2013, "$O_{2\,pre}$").

We believe that SS for "oxygen saturation state" may be quite immediate for readers (we changed this from $SS_{O2}$), same for "t0" for "at time 0" and "t" for "at time t". As for the difference between our notation and that used in Wakelin et al. 2020, we changed it because we believe the notation in Wakelin et al. 2020 may have been confusing (e.g. DOs for "oxygen saturation state", dimensionless, and DO for "oxygen concentration", concentration).

REV3:

"comment on the product rule" see online

ANSWER:

Reviewer 3 is correct in pointing out that "$\Delta O_{2,other}$ arbitrarily combines a 1st-order term with the 2nd-order term" and that the second order contribution, $\Delta f \times \Delta[O2]^{sat}$, should be quantified separately. We modified the methods section, and the results, were relevant, to account for this more rigorous approach. This includes computing the second order term.

However, the main body of results are not based on the $\Delta O_2$ decomposition, but on correlations between $[O2]^{sat}$ and f with other variables, and it can be demonstrated how, given any variable X,

$corr(X, f0 \times \Delta[O2]^{sat}) = corr(X, [O2]^{sat})$, and

$corr(X, \Delta f \times [O2]^{sat}_0) = corr(X, f)$

hence the main bulk of our results still stands. We however amended the description of the decomposition of $\Delta[O2]$ in the methods and presented the separate contribution of the second order term in the results. The contribution of the second order term turned out to be negligible.

---

## Referee Report (RR1)

**egusphere-2023-1049-manuscript-version2 Review**

**Title**: Multi-model comparison of trends and controls of near-bed oxygen concentration on the Northwest European Continental Shelf under climate change

This is my second review of this manuscript. On my first review I had recommended rejection mostly because a major conclusion was drawn from a correlation that was taken as if it was causal without justification. The authors have addressed this point fairly well in their response by clarifying that the water column equilibrates with atmospheric $O_2$ on short timescales mostly because of the strong mixing in the region, even if only seasonal. This is a key assumption and thus a welcome addition to the methods and the discussions. I would only argue that the main mechanism that degrades the causal link between $O_2$ and saturated $O_2$ is mixing of different waters (that can come from surface regions of different saturation states, e.g., near sea-ice or in upwelling regions) rather than the water age, although one could argue that age correlates well with mixing of diverse surface sources.

That being said, I think the manuscript in its current form requires major revisions before publication. The arguments are a little scattered and hard to follow, the decomposition methodology is a little confusing, there is a little too much information to digest in the form of complex spatial correlations (all of which have to be carefully inspected to determine if they represent causality or coincidence), the figures require improvements, and the main text remains filled with typos and incorrect capitalization/punctuation. Below I provide some major points and minor points for the authors to consider.

**Major points:**

- The structure of the paper (mostly the results section) could be improved. I would recommend starting with the oxygen changes, which is the main focus of this study (i.e., move 3.3 to 3.1). Section 3.1 (Validation) could be relegated to an appendix or to Section 4 to discuss the reliability of the results. Section 3.2 ($\Delta T$ and $\Delta S$) could be placed later when these variables are invoked to explain different mechanisms and correlations. Section 3.4 (contributions to $O_2$ change) seems that it should include 3.5 (contributions from T and circulation) and 3.6 (contributions from biology). The following Section (3.7; Impact of abrupt changes in circulation) seems a little off-beat given that it is the only section supported by a time series (Fig. 10). Maybe the conclusions from Fig. 10 can be presented also in $\Delta$'s that match all the previous results/figures?

- I remain unconvinced that the decomposition of the authors of $O_2 = O_2{}^{sat} \times SS$ is more useful than the traditional $O_2 = O_2{}^{sat} - AOU$. In the revised manuscript and to the other referees pointing to this in their first review, the authors responded that their method is different in that it focuses on change with respect to a reference period. However, this is entirely doable with AOU as well, simply through $\Delta O_2 = \Delta O_2{}^{sat} - \Delta AOU$. It thus appears that the SS decomposition only makes the paper unnecessarily convoluted.

- To reduce the number of panels, shorten the paper, and clarify which features/mechanisms are robust across models, maybe the authors could merge some panels, as is commonly done in CMIP studies? For example, Figure 4 in Busecke et al. (2022; doi: 10.1029/2021AV000470) uses dots to indicate where most of the models disagree on the sign of the 2000–2100 $O_2$ trend in the Pacific

OMZ. In a similar vein, to lend a helping hand to the reader, maybe the authors could use a distinct overlay/hash to indicate where they think the correlations are not to be understood as causal. Overall I think the paper would benefit from summarizing the Figures visually.

- Given that the authors focus on near-bed oxygen and thus benthic ecosystems, it might be good to consider changes in $pO2$ rather than $O_2$ concentrations (as advocated by, e.g., Seibel (2011; doi:10.1242/jeb.049171) and Hofmann et al. (2011; doi:10.1016/j.dsr.2011.09.004)). Better yet might be to consider some metabolic index, e.g., such as the one by Deutsch et al. (2015; already cited by the authors), although that might arguably be out of the scope of this work. Importantly however, the authors should discuss the temperature dependence of the tolerance of benthic organisms to reduced $O_2$ (e.g., Deutsch et al., 2020; doi:10.1038/s41586-020-2721-y), which might exacerbate the impact of deoxygenation on benthic ecosystems.

**Minor points:**

- "ecosystem" can be replaced by "biological" in many places for clarity.
- Many long multiple-idea sentences could be split up
- Avoid switching between "variables" and "parameters" if possible.
- Avoid the use of "common to X and Y" and instead maybe use "the same in X and Y"
- "Changes in $\Delta X$" is incorrect. It's either just "$\Delta X$" or "changes in X".
- What exactly is the correlation shown in most figures? Over what is it computed? Over the time periods? Both other referees requested equations in the previous review but only some quite unclear text was added.
- Minus signs should be proper minus signs "$-$" if possible (instead of hyphens "-")
- Sentences starting without a capital letter should be fixed.
- Random capitals mid-sentence should be fixed.
- Typos persist in this revised version.

1. Introduction:

    - L65: What are example of sub-lethal effects? I think one could be vision loss (e.g., McCormick et al. (2017; doi:10.1098/rsta.2016.0322)) but maybe the authors had other effects in mind that they should explicitly list here.
    - L70-74: Simplify to 2 significant digits and use the same unit (all in % or all in concentration) for clarity.

2. Methods:

    - Fig. 1: Add circulation arrows to guide the reader through the region dynamics if possible.
    - L180: Explain what climate sensitivity means: After how many years of 2×pCO$_2$ is the change in $T$ given?
    - L190: Explain what the version differences mean. What has changed between them?
    - L191: Explain what do the functional types difference applies to and what these differences are.
    - L206: the ocean color data product needs a reference.
    - L209: What does "setting low parameter values" do? Which parameters?
    - L209: What is "climatological" used for here? I think the authors mean "forced by climatological mean observations". Models can be deemed climatological too.
    - L212: Space after dot is missing.

- L216: That the nitrogen deposition field was "downloaded 2011" is not useful. Give a reference instead.
- L216: Anything special or descriptive can be said about tidal forcing? Why two citations and no explanation?
- L219: Is the "zero-gradient scheme" what is commonly known as Dirichlet boundary condition? If so name it that way.
- L323–239: Rewrite nbias and nurmsd paragraph, which is currently obscure and repetitive. An equation for each term would not hurt, as suggested by the other referees before. Using equations and less text can be good for clarity and brevity.
- L244: The parenthetical is unclear: Enhanced stratification does not limit atmospheric oxygen uptake, at least not on the regional scale, and Changes in circulation include changes in lateral transport by definition.
- L249+: What about "works as an approximation" instead of just "works". Also, what about "saturated" instead of "relaxed": AOU assumes complete saturation. Assumptions about it are not "change a little" but they are "does not change" instead.
- Eqs. (1) and (2) are not useful in my opinion. Add an Equation for $O_2 = SS \times O_2^{sat}$ instead, if you must. Related: Maybe I missed it, but how are $O_2^{sat}$ and SS computed? Is $O_2^{sat}$ an explicit tracer in the models? Is it computed directly from atmospheric $pO_2$ and in situ $T$ and $S$?
- L269: The "discrete product rule" is not really a thing, although I guess it could be. (This is my fault for naming it that way, thinking it made sense as a comment. The "product rule" is a thing, but that's not what the authors are using.) Either way, this is basic calculus that does not need a name, so what about simply: "Oxygen change between t0 and t can be decomposed as follows:"
- L274: replace "being $SS_t$" with "$SS_t$ being"
- Remove Eqs. (4) to (6), and add braces below Eq. (3) terms instead.
- L297+: This false-positives part is a little obscure to me. Can the authors simplify it?
- L305: Replace "$O_2^{sat}$ / AOU" with "$O_2 = O_2^{sat}$ - AOU" to avoid confusion. ("/" can mean "divided by")
- L306: The difference with AOU is not "the reference period". See major point.
- Make it clear here that Δsolubility captures most of the change in $O_2^{sat}$ on the shelf because here intense vertical mixing dominates open ocean contributions.

3. Results

  1. Ensemble validation:
     - So what? What is over/underestimated?
     - Delegate to appendix or discussion.
     - Fig. 2:
       - Colors would be welcome.
       - Add what is optimal/best in the caption. Is it (1,0), (0,0), or something else?
       - Use words and function names in parentheses in the caption.
       - Row labels are missing (I am guessing they are the 3 models)
       - Maybe bad suggestion: since these are normalized metrics, the axes could be shared and only be shown on the left for the y-axis of the left-most panels and the

bottom for the x-axis of the bottom-most panels (and the "cross" at (0,0) could be shown without the values for tick labels).

2. Changes in temperature and salinity
   - Fig. 3:
     - (Also applies to most maps) permuting the layout would allow for bigger panels and avoid requiring the reader to zoom in.
     - (Also applies to most maps) units could be better placed near the colorbars rather than in the title.
     - (Also applies to most maps) Discrete colormaps and filled contours could help for humans to extract values and visualize fronts
     - Show past and future $T$ and $S$ too in appendix/supplement?
3. Near-bed oxygen current state and change
   - Fig. 4:
     - Show future $O_2$?
     - What are the red spots when zooming in?
     - Do the high hypoxia incidence coincide with the highest past $O_2$ levels? Is this meaningful to discuss?
     - Is there no hotspot $O_2$ decline for GFDL? What thresholds define hotspots?
4. Contributions to near-bed $O_2$
   - L364: What is "negligible"? 10%? 1%? Less? It is important to be precise and quantify these terms because they are nonlinear (sometimes quadratic or worse), such that if they start gaining momentum as the climate changes, there is a chance they become dominant eventually.
   - L376: $\Delta SS_{O2}$ notation unused elsewhere.
   - Fig. 5: Colorbar tick labels of last column are rounded too aggressively.
5. Physical controls of $\Delta O_2$: temperature and stratification
   - L382: "Changes in $\Delta O_{phy-ch}$" does not work.
   - Fig. 6: Why does the white turn gray for this figure?
   - L397: missing punctuation before "de Boer"
   - L401: Too many "is" in the sentence.
   - L404: Replace "mediated" by "caused"
6. Biogeochemical controls of oxygen change: primary production and respiration
   - L414: Not "all models": $\Delta NPP$ looks to be positive for HADGEM (more strong red).
7. Impact of abrupt changes in circulation on the emergence of deoxygenation hotspots
   - L452: What are "R" and "p"?
   - Fig. 10: panel labels are not consistent with previous figures, which have no ending parenthesis, e.g., "a" vs "a)".

4. Discussion

- L465: Odd space
- L466: Move Holt et al. reference to just before the comma.
- L494: Remove "by critical hypoxia" (it is clear that you are talking about hypoxia for which you just defined the threshold)
- L493: "Oschlies" is misspelled.
- L533: Remove end of sentence: "testifying (...) in our ensemble" (redundant).

- L536: Capitalize RCP and define it (and cite appropriate reference)

5. Conclusions

- L571: There is no "World" in CMIP.

---

## Referee Report (RR2)

**egusphere-2023-1049-manuscript-version3 Review**

**Title**: Multi-model comparison of trends and controls of near-bed oxygen concentration on the Northwest European Continental Shelf under climate change

This is my third review of this manuscript. In my previous review I recommended major revisions related to the paper's structure, the methodology, the figures, and other minor issues. The structure and the figures have been greatly improved, making the paper easier to follow and straighter to the point. The explanations are clearer and the conclusions better supported. Overall I think the manuscript now only needs minor adjustments before publication.

My biggest concern relates to the justification for the authors' approach for decomposing oxygen (they use $O_2 = SS \times O_{2,sat}$) and to its comparison with the "AOU approach" ($O_2 = O_{2,sat} - AOU$). In their last response and revision, the authors have added claims of advantages over the AOU approach, which are not supported, and some confusing/incorrect statements remain regarding the differences between the approaches and about the assumptions underlying the AOU approach. I emphasize that I am not suggesting the authors should change their methodology. It is entirely up to them to choose the approach they think is best suited to support their conclusions, even if that choice is arbitrary. However, unsupported claims and incorrect statements should be fixed or removed. Below I detail the issues relating to the authors approach, and then list some minor points and suggestions, which I think will require only minor revisions, hence my recommendation.

**Authors approach vs AOU approach**

1. In their response, the authors suggest that an advantage of their approach is that it allows to cleanly separate the solubility-only contribution while the AOU approach does not because $\Delta AOU$ contains a $\Delta O_{2,sat}$ term in (eq. S8):

   > $\Delta AOU = (1 - SS_{t0}) \Delta O_{2,sat} - O_{2,sat,t0} \Delta SS.$

   There is a logical and a numerical issue with this argument:

   1. The logical issue is that of circular reasoning, in that the argument is based on the implicit premise that $\Delta SS$ is independent of $\Delta O_{2,sat}$ in the first place. In fact, one could apply the same argument to $\Delta SS$ just as well. That is, one could write $\Delta SS$ as a function of $\Delta AOU$ and $\Delta O_{2,sat}$ through the Taylor expansion of $SS = 1 - AOU / O_{2,sat}$, and then argue that $\Delta SS$ contains $\Delta O_{2,sat}$ terms.

   2. The numerical issue is that even if the premise (that $\Delta SS$ is independent of $\Delta O_{2,sat}$) were true, the detailed proof in the supplement is not (yet) supported by the data. In the proof, the authors argue that the $\Delta O_{2,sat}$ dependency would only disappear if $SS_{t0} = 1$ and that this condition is not verified in their case. However, $1 - SS_{t0}$ need not be zero. It just needs to be small enough (maybe $SS_{t0} \gtrsim 90\%$ suffices). If so, the $(1 - SS_{t0}) \Delta O_{2,sat}$ term would be of second order and could be discarded just as the "mix" term. Figures showing the magnitude of $SS$ (past and future values, not the change $\Delta SS$) may help prove/disprove this point.

I would recommend removing these claims and arguments entirely, unless the authors can provide a proof that $\Delta SS$ is independent of $\Delta O_{2,\text{sat}}$.

2. The difference between the authors' approach and the AOU approach is incorrectly stated in the main text, where the authors have essentially kept the original statement suggesting that the difference lies within "accounting for a reference period", which I had already pointed as incorrect in my previous review. The reference period is separate and is not the difference between the two approaches. In fact, the authors themselves, in the supplement, use the same reference period for both approaches. I would thus suggest to not mention the "reference period" when discussing methodological differences with the AOU approach.

3. There is a recurring confusion between the condition of complete surface saturation ($SS_t = 1$ for all $t$ at the surface, the assumption for AOU) with the condition of complete saturation initially everywhere ($SS_{t0} = 1$, which comes from the circular argument; point 1. above). This recurring confusion appears in both words and equations in the authors' response (to Reviewer 3, 2nd review, Major comment 2), in the main text (L270), and in the supplement (L102–103). This is strange because the assumption for AOU (surface saturation) is correctly stated elsewhere in the main text (L237) and even right after the incorrect assumption in the supplement, where the authors further suggest that they are the same (supplement L103–104):

> (...) under the assumption that $SS_{t0} = 1$. This is in line with Duteil et al. (2013) that assumed saturation was reached at surface.

I would suggest removing any mention of this odd "initial saturation ($SS_{t0} = 1$)" condition which is not relevant to AOU.

**Minor points/suggestions:**

- L78:

  > whilst away from the euphotic zone respiration exceeds primary production, resulting in net oxygen consumption.

  Is there **any** primary production away from the euphotic zone? If not, what about "whilst away from the euphotic zone respiration removes oxygen."

- L83: Replace "limit" with "reduce" for clarity

- L270:

  > The metrics in eq. 2 and 3 are related to the classic $O_2 = O_{2,\text{sat}} -$ AOU decomposition, with the difference that, by explicitly accounting for a reference period, they relax the AOU assumption of complete saturation at t0.

  This is incorrect on two fronts: the difference does not lie in the reference time t0, and the AOU assumption is not complete saturation at t0 (see major point above).

- Eq. (4): Note that "mix" as a subscript for the cross term could be confused for the contribution from (water) mixing.

- L274: Add comma

  > To assess what drives the oxygen changes we computed (...)

  like so

  > To assess what drives the oxygen changes, we computed (...)

  otherwise it may read as "the oxygen changes (that) we computed".

- L296: use minus signs instead of hyphens "−0.5 mg L$^{-1}$" (also in the exponent)

- L302: missing minus sign in exponent in "6 mg L$^{-1}$"

- L327: missing space after "Fig 3d)"

- L331: extra closing parenthesis in "(Fig. 4g-l))"

- L333: missing "the" before "Central North Sea"

- L375: missing space after "(Fig. 6d, e, f)."

- Fig. 7 caption: Capitalize 1st letter of 2nd sentence.

- L408:

  > (...) the correlation between SS and BResp is rather weak and, at some locations, positive (instead of negative as would be expected, Fig. 8e, f). This is because simple point-to-point correlation over the full period, does not allow to capture seasonally heterogeneous process.

  Another reason could be that the point-to-point (in both time and space) correlation cannot capture the effect on SS from the distant respiration that occured in the past and at upstream locations from where the SS is computed.

- L450: replace comma after "etc" with dot

**Previous points where feedback was requested or required**

- Sorry for this unclear comment:

  > Use words and function names in parentheses in the caption.

  I assumed "nbias" and "nurmsd" were function names but maybe these are just acronyms? Anyway, what I intended to suggest was to use, e.g.:

  > Fig. S1. Validation results. Plots show normalised bias (nbias) vs normalised unbiased root mean squared (nurmsd) for selected variables in the three ensemble members and in different model subdomains. A perfect fit would sit at the origin (0.0, 0.0).

- Fig. 2 (formerly Fig. 3) comment:

  > Show past and future $T$ and $S$ too in appendix/supplement?

Sorry for not being explicit enough. I did not mean for the authors to move or remove this figure. Instead, I meant to suggest **_adding_** a supplement figure of past and future $T$ and $S$ to provide additional information and context to this $\Delta T$ and $\Delta S$ figure.

- Fig. 3 (formerly Fig. 4) comment:

> Show future $O_2$?

As point above there I also wanted to suggest **_adding_** plots of future $O_2$ without removing $\Delta O_2$.

---

## Author Response (AR2)

**Reviewer 1**

**Comment**
In the introduction (line 85) and the discussion (line 498), the authors remark on oxygen minimum zones. This seems to me unnecessary, and possibly confusing, as the study region is not in or near an oxygen minimum zone.
**Answer**
We agree these two sentences may have been misleading as the NWES is not an oxygen minimum zone. Here we wanted to point out how the processes that shape large scale oxygen minimum zones are similar in smaller scale low oxygen areas. We clarified this in the text.

**Comment**
At some point, the abbreviation in $O_{2,phy-ch}$ should be explained. The term 'physico-chemical' does not appear in the text (I think) in the revised manuscript.
**Answer**
We explained the abbreviation phy-ch, physico-chemical, in the methods.

**Comment**
Ensemble members should be labeled in Fig. 2
**Answer**
We added ensemble labels to fig. 2. (NB the figure is now moved to an appendix after reviewer 3 suggestion).

**Comment**
As a general comment, the prose throughout the text are often a bit wordy or conversational. This can make the point hard to follow at times. The text would benefit from some simplification of sentence structure where possible.
**Answer**
We carefully read the paper and took care to improve on the style and simplify convoluted constructions as much as possible, especially in the results section.

**Reviewer3**

We thank the reviewer for the useful comments. We implemented the suggested changes as detailed below.
In particular, we took care of improving the justification of our methodology, we simplified and added clarity to the results exposition, and we moved some information to a supplement.

**Major points**

**Comment:**
The structure of the paper (mostly the results section) could be improved. I would recommend starting with the oxygen changes, which is the main focus of this study (i.e., move 3.3 to 3.1). Section 3.1 (Validation) could be relegated to an appendix or to Section 4 to discuss the reliability of the results. Section 3.2 ($\Delta T$ and $\Delta S$) could be placed later when these variables are invoked to explain different mechanisms and correlations. Section 3.4 (contributions to $O_2$ change) seems that it should include 3.5 (contributions from T and circulation) and 3.6 (contributions from biology). The following Section (3.7; Impact of abrupt changes in circulation) seems a little off-beat given that it is the only section supported by a time series (Fig. 10). Maybe the conclusions from Fig. 10 can be presented also in $\Delta$'s that match all the previous results/figures?
**Answer:**
We thank the reviewer for their suggestions to streamline the results section of the paper. As suggested, we moved the validation in the supplementary information. We still prefer to keep the order of the remaining sections as we originally envisaged, as this would allow to provide first a short summary of the impact of climate change on the NEWS T and S to provide the context in which oxygen changes will be analysed, and then continue to focus on oxygen for the rest of the section without any disruption. Similarly, we decide against accepting the suggesting in merging sub-sections 3.4, 3.5 and 3.6 (now 3.3, 3.4, and 3.5) as we believe that the current structure with sub-heading help the reader to navigate across the various aspect of the analysis.
We have added a sentence to highlight the rational at the beginning of the section.
Finally, regarding sub-section 3.7 (now 3.6), while we understand that being the only section where time series are presented might have initially puzzled the reviewer, we believe that showing maps of D of the circulation would be less informative than comparing the time evolution of the current with that of the oxygen change because the synchronous aspect of the changes would be missed.
However, we have changed figure 10 (now figure 9) showing D $O_{2,\text{sat}}$ and D SS instead of $O_{2,\text{sat}}$ and SS to be consistent with the other sections.

**Comment**
I remain unconvinced that the decomposition of the authors of $O_2 = O_2^{\text{sat}} \times$ SS is more useful than the traditional $O_2 = O_2^{\text{sat}}$ - AOU. In the revised manuscript and to the other referees pointing to this in their first review, the authors responded that their method is different in that it focuses on change with respect to a reference period. However, this is entirely doable with AOU as well, simply through $\Delta O_2 = \Delta O_2$ - $\Delta$AOU. It thus appears that the SS decomposition only makes the paper unnecessarily convoluted.
**Answer**
We understand how our $\Delta O_2$ decomposition needs additional support as also the other reviewers raised concerns about it. To answer these concerns we wrote down the equations to explicitly compare our $\Delta O_2$ decomposition to the one the reviewer suggests, $\Delta O_2 = \Delta O_{2,\text{sat}}$ - $\Delta$AOU. The full proof is in the supplementary material. In short, it can be demonstrated that, while the two decompositions are equivalent, $\Delta$AOU does depend on both $\Delta O_{2,\text{sat}}$ and $\Delta$SS (Eq S6).
This means $\Delta$AOU doesn't just account for $O_2$ changes due to transport and biology, but also for part of the changes due to changes in solubility. This is problematic as AOU is usually interpreted as

the component of $O_2$ change not due to changes in solubility. We then go on to prove that this discrepancy is reconciled if we assume complete saturation at the initial conditions ($SS_{t0} = 1$), which, as the reviewer pointed out, is an assumption of the AOU model. This assumption is not present in our decomposition, and this allow us to better separate the effect of changes in solubility from the one due to the other changes.

we thank the reviewer for suggesting to be more detailed in the description of the mathematical passages behind our assumption and analysis as we believe these could be helpful for readers seeking to make sense of some common practice of O2 change decomposition, that is given for granted in many studies.

**Comment**

To reduce the number of panels, shorten the paper, and clarify which features/mechanisms are robust across models, maybe the authors could merge some panels, as is commonly done in CMIP studies? For example, Figure 4 in Busecke et al. (2022; doi: 10.1029/2021AV000470) uses dots to indicate where most of the models disagree on the sign of the 2000–2100 O2 trend in the Pacific OMZ. In a similar vein, to lend a helping hand to the reader, maybe the authors could use a distinct overlay/hash to indicate where they think the correlations are not to be understood as causal. Overall I think the paper would benefit from summarizing the Figures visually.

**Answer**

We thank the reviewer for the suggestion about hatching areas of interest, and we have adopted that in figure 3 to highlight the hotspots of de-oxygenation, that we defined with rigorous thresholds. Unfortunately, it is more difficult for us to define a clear threshold for the correlation to be considered not causal, because that assessment is done not purely on a quantitative metric (the strength of the correlation) but also on the existence of a causal relationship. To help the reader, we always specify the name of the areas using the same toponym used in figure 1, so that readers not familiar with the area could still orient themselves.

Furthermore, we're not sure that merging the panels for each model in a single panel similarly to Busecke et al (2022) would be beneficial in this case. In that paper the authors analysed a much wider ensemble (14 models) and so they needed to give an indication of how spread the ensemble was. In our case, we only have 3 members, and therefore the ensemble is too small to make any such consideration. Besides, looking at each model separately allow to better understand the causal relationship behind the observed changes.

We however recognise the result section may have been at times hard to follow. We made several changes throughout the section to alleviate this. We carefully modified the text, simplified the exposition whenever possible and avoided unnecessary repetitions. We put extra care on highlighting the information we want to focus on, especially causal relations, and we added references to panel labels to guide the reader.

**Comment**

Given that the authors focus on near-bed oxygen and thus benthic ecosystems, it might be good to consider changes in pO2 rather than $O_2$ concentrations (as advocated by, e.g., Seibel (2011; doi:10.1242/jeb.049171) and Hofmann et al. (2011; doi:10.1016/j.dsr.2011.09.004)). Better yet might be to consider some metabolic index, e.g., such as the one by Deutsch et al. (2015; already cited by the authors), although that might arguably be out of the scope of this work. Importantly however, the authors should discuss the temperature dependence of the tolerance of benthic organisms to reduced O2 (e.g., Deutsch et al., 2020; doi:10.1038/s41586-020-2721-y), which might exacerbate the impact of deoxygenation on benthic ecosystems.

**Answer**

We agree pO2 or metabolic indexes (which are based on pO2 rather than oxygen concentration) like the ones in Deutsch et al. 2020 or Clarke et al. 2021, are more meaningful for impacts on organisms than oxygen concentration. As the reviewer points out, our paper focusses on deoxygenation rather

than on its impacts. We however agree this an important point to discuss and we added a dedicated paragraph at the very end of the discussion.

**Minor points**

**Comment**
"ecosystem" can be replaced by "biological" in many places for clarity.
**Answer**
We replaced "ecosystem" with "biological" where it was fit.

**Comment**
Many long multiple-idea sentences could be split up.
**Answer**
We carefully re-read the text, taking care of splitting up long sentences with full stops whenever possible.

**Comment**
Avoid switching between "variables" and "parameters" if possible.
**Answer**
We made sure that in the text "variables" always refers to model state or diagnostic variables and "parameters" always refers to model parameters.

**Comment**
Avoid the use of "common to X and Y" and instead maybe use "the same in X and Y"
**Answer**
We replaced all "common to" with "the same in".

**Comment**
"Changes in $\Delta X$" is incorrect. It's either just "$\Delta X$" or "changes in X".
**Answer**
We made sure not to use "change in $\Delta X$".

**Comment**
What exactly is the correlation shown in most figures? Over what is it computed? Over the time periods? Both other referees requested equations in the previous review but only some quite unclear text was added.
**Answer**
The Spearmann correlations are computed at each grid point between pairs of monthly averaged timeseries (so over time) for the whole time period. We modified the relevant text in the methods section to make this clearer. We don't feel we need to add an equation for Spearmann correlation, which is widely used.

**Comment**
Minus signs should be proper minus signs "−" if possible (instead of hyphens "-")
**Answer**

We replaced hyphens with minus signs where fit.

**Comment**
Sentences starting without a capital letter should be fixed. Random capitals mid-sentence should be fixed. Typos persist in this revised version.
**Answer**
We carefully re-read all the text and corrected typos and capitalisations.

**1. Introduction**

**Comment**
L65: What are example of sub-lethal effects? I think one could be vision loss (e.g., McCormick et al. (2017; doi:10.1098/rsta.2016.0322)) but maybe the authors had other effects in mind that they should explicitly list here.
**Answer**
We were thinking more generally about depression of metabolism with low oxygen levels, but also impaired vision is a good and more specific example. We added a reference for each.

**Comment**
L70-74: Simplify to 2 significant digits and use the same unit (all in % or all in concentration) for clarity.
**Answer**
We agree it would be advisable to provide this information with coherent units. Unfortunately the Bopp and Kwiatkowski papers provide this information with different units and it is not possible to convert one in the other based only on the final papers. The aim of this sentence is to set up the context of declining ocean oxygen and comparing different global estimates is outside the scope of this paper.

**2. Methods**

**Comment**
Fig. 1: Add circulation arrows to guide the reader through the region dynamics if possible.
**Answer**
We added circulation arrows to Fig. 1 and indicated the western Norwegian trench current in the caption.

**Comment**
L180: Explain what climate sensitivity means: After how many years of $2\times pCO_2$ is the change in T given?
**Answer**
The equilibrium climate sensitivity is estimated once a model run reaches stationary conditions (hence 'equilibrium') after an instantaneous doubling of atmospheric $pCO_2$. This may take different time in different models. We don't think a quantification of this is necessary to follow the rest of the paper.

**Comment**
L190: Explain what the version differences mean. What has changed between them?
**Answer**
Changes between NEMO versions typically include numerical schemes, the representation of physical processes, modules coupling, software and hardware related matters, domain

configurations, etc. All of these are quite complex topics and it is out of the scope of this study to thoroughly describe them. It wouldn't be useful either because it is not straightforward to attribute any of our results to changes in NEMO versions; and indeed we don't. We think here it suffices to refer to the relevant literature where these are described with more detail.

**Comment**
L191: Explain what do the functional types difference applies to and what these differences are.
**Answer**
Here a similar reasoning applies. The two ERSEM versions differ in many regards. Noticeably in Phytoplankton and Bacteria parameterisation, although this is not the only thing that changed. These changes probably do have some effect on our results (which we do mention in the discussion). However, attributing a certain result to the different ERSEM version is not straightforward and would require ad-hoc experiments, which we don't have time or resources to run. And we don't think that would add anything key to our results either. We think it should suffice here to cite the relevant literature where the different ERSEM versions are described in detail.

**Comment**
L206: the ocean color data product needs a reference.
**Answer**
We added a reference for the Ocean Colour product.

**Comment**
L209: What does "setting low parameter values" do? Which parameters?
**Answer**
By setting biogeochemical model state variables to low values we mean positive but close enough to zero to be negligible (e.g. 1e-5). This is because most model biogeochemical state variables cannot be negative or =0. We replaced "low" with "close to zero" in the text.

**Comment**
L209: What is "climatological" used for here? I think the authors mean "forced by climatological mean observations". Models can be deemed climatological too.
**Answer**
We replaced "climatological" with "forced with climatological mean values". Note that while Baltic boundary biogeochemistry is from observations, the physics are from a reanalysis product. This is specified in the following sentence.

**Comment**
L212: Space after dot is missing.
**Answer**
A space was added after the dot.

**Comment**
L216: That the nitrogen deposition field was "downloaded 2011" is not useful. Give a reference instead.
**Answer**
We added a reference for the N deposition product.

**Comment**
L216: Anything special or descriptive can be said about tidal forcing? Why two citations and no explanation?

**Answer**

Tidal mixing is recognised as an important process in the NWES and tides are indeed routinely applied in all recent NWES regional ocean model runs we are aware of. However, we think a comprehensive description of tides implementation would not just be lengthy, but also out of the scope of this paper because it would't add anything substantial to the interpretation of our results.

**Comment**

L219: Is the "zero-gradient scheme" what is commonly known as Dirichlet boundary condition? If so name it that way.

**Answer**

The boundary scheme we use is not the Dirichlet one but a special case of the Neumann one where the gradient between each boundary cell and the adjacent interior cell is zero (hence zero-gradient). We think the sentence "the concentration at the boundary equals the concentration immediately inside the domain" should clarify this for readers.

**Comment**

L323–239: Rewrite nbias and nurmsd paragraph, which is currently obscure and repetitive. An equation for each term would not hurt, as suggested by the other referees before. Using equations and less text can be good for clarity and brevity.

**Answer**

We re-wrote the validation paragraphs (both the methods and results), also adding equations, and moved them to supplementary material, as suggested.

**Comment**

L244: The parenthetical is unclear: Enhanced stratification does not limit atmospheric oxygen uptake, at least not on the regional scale, and Changes in circulation include changes in lateral transport by definition.

**Answer**

Agree, we changed "atmospheric oxygen uptake" to "vertical transport" (i.e. from the mixed layer to deeper layers), but retained the "lateral transport" bit because it is indeed changes in ocean circulation that produce (also) changes in lateral oxygen transport.

**Comment**

L249+: What about "works as an approximation" instead of just "works". Also, what about "saturated" instead of "relaxed": AOU assumes complete saturation. Assumptions about it are not "change a little" but they are "does not change" instead.

**Answer**

Agree, we implemented the suggested changes.

**Comment**

Eqs. (1) and (2) are not useful in my opinion. Add an Equation for $O_2 = SS \times O_{2,sat}$ instead, if you must. Related: Maybe I missed it, but how are $O_2$ and SS computed? Is $O_{2,sat}$ an explicit tracer in the models? Is it computed directly from atmospheric $pO_2$ and in situ T and S?

**Answer**

We removed equations (1) and (2). $O_{2,sat}$ is computed (at runtime by ERSEM) from temperature and salinity according to Weiss 1970 (atmospheric $pO_2$ is constant), $SS = O_2 / O_{2,sat}$. We added these informations in the text.

**Comment**

L269: The "discrete product rule" is not really a thing, although I guess it could be. (This is my fault for naming it that way, thinking it made sense as a comment. The "product rule" is a thing, but that's

not what the authors are using). Either way, this is basic calculus that does not need a name, so what about simply: "Oxygen change between t0 and t can be decomposed as follows:"
**Answer**
We removed "by the discrete product rule".

**Comment**
L274: replace "being $SS_t$" with "$SS_t$ being"
**Answer**
We agree, done.

**Comment**
Remove Eqs. (4) to (6), and add braces below Eq. (3) terms instead.
**Answer**
We think that adding braces below Eq. (1) terms may end up not looking too good on a typical 2-column paper layout. We would prefer to keep eq 2, 3 and 4 separated.

**Comment**
L297+: This false-positives part is a little obscure to me. Can the authors simplify it?
**Answer**
We re-wrote the paragraph, we think it is much clearer now.

**Comment**
L305: Replace "$O_2$ / AOU" with "$O_2 = O_2$ - AOU" to avoid confusion. ("/" can mean "divided by")
**Answer**
We rephrased the decomposition as $O_2 = O_{2,sat} - AOU$

**Comment**
L306: The difference with AOU is not "the reference period". See major point.
**Answer**
As mentioned earlier, we added in the supplementary material a proof of how our decomposition differs from an AOU-based one. The difference is that our decomposition does not assume complete saturation at t0. We added this information in the text.

**Comment**
Make it clear here that Δsolubility captures most of the change in $O_2$ on the shelf because here intense vertical mixing dominates open ocean contributions.
**Answer**
We added a mention to the fact that vertical mixing dominates open ocean exchange.

**3. Results**

**3.1 Ensemble Validation**

**Comment**
So what? What is over/underestimated?
**Answer**

As mentioned, we re-wrote the validation sections and moved them to supplementary material, as requested. Among other changes we took care of explicitly stating what is over- and under-estimated in our ensemble members.

**Comment**
Delegate to appendix or discussion.
**Answer**
We moved validation to the supplementary material.

**Fig. 2:**

**Comment**
Colors would be welcome.
**Answer**
We had some trials adding colours to the validation figure. Unfortunately this doesn't help and actually makes things worse when symbols overlap, making them very hard to tell apart. We think however that the figure is still effective in summarising the validation results. Even when symbols are clustered, that indicates there's not much significant differences among them.

**Comment**
Add what is optimal/best in the caption. Is it (1,0), (0,0), or something else?
**Answer**
a perfect fit would be (0,0), i.e. 0 bias and 0 rmsd. We added this information in the caption.

**Comment**
Use words and function names in parentheses in the caption.
**Answer**
Unfortunately we don't understand what words and function names should go in parentheses. As we don't use parentheses for function names anywhere in the text, we don't understand why we should do it here. We are sure there's a point here. Unfortunately we struggle to understand it. We are happy to follow the reviewer suggestion once clarified.

**Comment**
Row labels are missing (I am guessing they are the 3 models).
**Answer**
We added the row labels

**Comment**
Maybe bad suggestion: since these are normalized metrics, the axes could be shared and only be shown on the left for the y-axis of the left-most panels and the bottom for the x-axis of the bottom-most panels (and the "cross" at (0,0) could be shown without the values for tick labels).
**Answer**
As the reviewer anticipates, using the same axes limits for all plots here is not advisable as it would further cluster together the sub-basin labels in most plots, making them unreadable.

**2. Changes in temperature and salinty**

**Fig. 3**

**Comment**
(Also applies to most maps) permuting the layout would allow for bigger panels and avoid requiring the reader to zoom in.
**Answer**
Please note that the figure size in the manuscript are not the actual final figure sizes. These will ultimately be up to the editors of the journal, however we sized our figures so that a 2-column 3-rows figure like Fig 2 has the width of one column in a typical 2-column paper layout (about 8.1 cm). With this figure size and with the appropriate resolution all panels will be clearly readable in the published paper.

**Comment**
(Also applies to most maps) units could be better placed near the colorbars rather than in the title.
**Answer**
Given how we structured the figures' layout, placing the units near the colorbar would come at the cost of having larger figures, without noticeable benefits for readibility.

**Comment**
(Also applies to most maps) Discrete colormaps and filled contours could help for humans to extract values and visualize fronts.
**Answer**
We replaced all maps (except the bathymetry in Fig. 1) with filled contours and discrete colormap ones.

**Comment**
Show past and future T and S too in appendix/supplement?
**Answer**
Changes in physics variables T and S is what ultimately drives changes in stratification, circulation and biogeochemistry, including oxygen. We think T and S are an important feature to show in the main text.

**3. Near-bed oxygen current state and change**

**Fig. 4**

**Comment**
Show future $O_2$?
**Answer**
Here we deemed more appropriate to show $\Delta O_2$, rather than future $O_2$, because the $\Delta$ is not affected by model bias.

**Comment**
What are the red spots when zooming in?
**Answer**
The red spots were an artefacts from the bias correction due to the regridding of the observation data. We removed them.

**Comment**
Do the high hypoxia incidence coincide with the highest past $O_2$ levels? Is this meaningful to discuss?
**Answer**
This is indeed an important point to underline. In short, these areas have high productivity (high $O_2$ production) and stratify seasonally (hence vulnerable to hypoxia). The study focusses on changes in average values rather than extremes, so we didn't include a full diagnosis of this. However, we added some lines in the results and in the discussion.

**Comment**
Is there no hotspot $O_2$ decline for GFDL? What thresholds define hotspots?
**Answer**
We added a definition of hotspot for oxygen decline as areas with $\Delta O_2 < -0.5$ mg $L^{-1}$ and $\Delta O_2 < 1.5$ $\Delta O_{2,mean}$, with $\Delta O_{2,mean}$ the average $\Delta O_2$ over the shelf. This information was included also in Fig. 3 where hotspots of $O_2$ decline are now hatched.

**4. Contribution to near-bed O2**

**Comment**
L364: What is "negligible"? 10%? 1%? Less? It is important to be precise and quantify these terms because they are nonlinear (sometimes quadratic or worse), such that if they start gaining momentum as the climate changes, there is a chance they become dominant eventually.
**Answer**
$\Delta O_{2,mix}$ in the last 30y of simulation is on average < 1% of total change in all members. We added this information in the text.

**Comment**
L376: $\Delta SS_{O2}$ notation unused elsewhere.
**Answer**
We corrected the typo.

**Comment**
Fig. 5: Colorbar tick labels of last column are rounded too aggressively.
**Answer**
We corrected the colorbar tick labels.

**5. Physical controls of ΔO2: temperature and stratification**

**Comment**
L382: "Changes in $\Delta O_{phy-ch}$" does not work.
**Answer**
Changed to "Changes in $O_{2,phy-ch}$".

**Comment**
Fig. 6: Why does the white turn gray for this figure?
**Answer**
Grey areas on shelf are where the correlation is non-significant. We added this information in the captions.

**Comment**
L397: missing punctuation before "de Boer"
**Answer**
We added a comma.

**Comment**
L401: Too many "is" in the sentence.
**Answer**
We removed "is" before "surface salinity".

**Comment**
L404: Replace "mediated" by "caused"
**Answer**
We replaced "mediated" with "caused".

**6. Biogeochemical controls of oxygen change: primary production and respiration**

**Comment**
L414: Not "all models": ΔNPP looks to be positive for HADGEM (more strong red).
**Answer**
Agree, we added this to the results description.

**7. Impact of abrupt changes in circulation on the emergence of deoxygenation hotspots**

**Comment**
L452: What are "R" and "p"?
**Answer**
We defined R and p in the methods. They are Spearman correlation coefficient (R) and p-value (p).

**Comment**
Fig. 10: panel labels are not consistent with previous figures, which have no ending parenthesis, e.g., "a" vs "a)".
**Answer**
We changed the panel labels.

**4. Discussion**

**Comment**
L465: Odd space
**Answer**
Odd space removed.

**Comment**
L466: Move Holt et al. reference to just before the comma.
**Answer**
Moved Holt et al. reference to after "changes" (because also Wakelin et al. reference is about the NWES).

**Comment**
L494: Remove "by critical hypoxia" (it is clear that you are talking about hypoxia for which you just defined the threshold)
**Answer**
Removed "by critical hypoxia".

**Comment**
L493: "Oschlies" is misspelled.
**Answer**
Corrected "Oschlies".

**Comment**
L533: Remove end of sentence: "testifying (...) in our ensemble" (redundant).
**Answer**
Removed "in our ensemble".

**Comment**
L536: Capitalize RCP and define it (and cite appropriate reference)
**Answer**
We capitalised RCP8.5. We would avoid, if possible, lengthy description of what Representative Concentration Pathways (RCP) and Shared Socioeconomic Pathways (SSP) scenarios are, and how they are defined, as this is common knowledge in climate science and it risks to unnecessarily weigh down the text.

**5. Conclusions**

**Comment**
L571: There is no "World" in CMIP.
**Answer**
Removed "World".

---

## Author Response (AR3)

**Rev3:**

This is my third review of this manuscript. In my previous review I recommended major revisions related to the paper's structure, the methodology, the figures, and other minor issues. The structure and the figures have been greatly improved, making the paper easier to follow and straighter to the point. The explanations are clearer and the conclusions better supported. Overall I think the manuscript now only needs minor adjustments before publication.

My biggest concern relates to the justification for the authors' approach for decomposing oxygen (they use $O_2 = SS \times O_{2,sat}$) and to its comparison with the "AOU approach" ($O_2 = O_{2,sat} - AOU$). In their last response and revision, the authors have added claims of advantages over the AOU approach, which are not supported, and some confusing/incorrect statements remain regarding the differences between the approaches and about the assumptions underlying the AOU approach. I emphasize that I am not suggesting the authors should change their methodology. It is entirely up to them to choose the approach they think is best suited to support their conclusions, even if that choice is arbitrary. However, unsupported claims and incorrect statements should be fixed or removed. Below I detail the issues relating to the authors approach, and then list some minor points and suggestions, which I think will require only minor revisions, hence my recommendation.

**Answer:**

We thank the reviewer for the useful comments, we have implemented all the suggested changed. In particular we amended both the methods section and the supplement and we no longer claim advantages of our approach over AOU. We also amended and/or removed all passages that were critical regarding the differences of the to approaches. All changes are detailed in the following.

We appreciate that the reviewer has not suggested to change the methodology, that we believe has similar validity of the more traditional AOU approach. We accept the reviewer that the difference between the two approaches is smaller than we implied and therefore we have decided to remove the final part of the supplementary information (from line100 onwards) and add the following:
*"In this region, the first term ((1 − SSt0) ΔO2,sat) is usually small at the annual scale (figure S2), and therefore the two approaches are largely equivalent."*

**Authors approach vs AOU approach.**

**Rev3:**

1. In their response, the authors suggest that an advantage of their approach is that it allows to cleanly separate the solubility-only contribution while the AOU approach does not because $\Delta AOU$ contains a $\Delta O_{2,sat}$ term in (eq. S8):

$$\Delta AOU = (1 - SS_{t0}) \Delta O_{2,sat} - O_{2,sat,t0} \Delta SS.$$

There is a logical and a numerical issue with this argument:

1. The logical issue is that of circular reasoning, in that the argument is based on the implicit premise that $\Delta SS$ is independent of $\Delta O_{2,sat}$ in the first place. In fact, one could apply the same argument to $\Delta SS$ just as well. That is, one could write $\Delta SS$ as a function of $\Delta AOU$ and $\Delta O_{2,sat}$ through the Taylor expansion of $SS = 1 - AOU / O_{2,sat}$, and then argue that $\Delta SS$ contains $\Delta O_{2,sat}$ terms.

**Answer:**

The reviewer is right in pointing this out. $\Delta SS$ and $\Delta O_{2,sat}$ are indeed assumed to be independent. We added one sentence in the methods explicitly stating the assumption of independence. This can be justified in our case study with the same argument that justifies the choice of the method for our

case-study. The timescales at which $O_2$ equilibrates with atmospheric concentration are short (due to intense wind and tidally-driven mixing) and ocean-atmosphere exchange dominates over shelf-ocean exchange. This effectively decouples the dynamics of $O_{2,sat}$ and SS, making them, as a first approximation, independent.

**Rev3:**
2. The numerical issue is that even if the premise (that $\Delta SS$ is independent of $\Delta O_{2,sat}$) were true, the detailed proof in the supplement is not (yet) supported by the data. In the proof, the authors argue that the $\Delta O_{2,sat}$ dependency would only disappear if $SS_{t0} = 1$ and that this condition is not verified in their case. However, $1 - SS_{t0}$ need not be zero. It just needs to be small enough (maybe $SS_{t0}$ 90% suffices). If so, the $(1 - SS_{t0}) \Delta O_{2,sat}$ term would be of second order and could be discarded just as the "mix" term. Figures showing the magnitude of SS (past and future values, not the change $\Delta SS$) may help prove/disprove this point.
I would recommend removing these claims and arguments entirely, unless the authors can provide a proof that $\Delta SS$ is independent of $\Delta O_{2,sat}$.

**Answer:**
The reviewer is indeed right in pointing this out. We have calculated the $(1 - SS_{t0}) \Delta O_{2,sat}$ term and it is indeed much smaller than the $O_{2,sat,t0} \Delta SS$ term. We added the plot to the supplement (past and future SS plots are also in the supplement).

**Rev3:**
3. The difference between the authors' approach and the AOU approach is incorrectly stated in the main text, where the authors have essentially kept the original statement suggesting that the difference lies within "accounting for a reference period", which I had already pointed as incorrect in my previous review. The reference period is separate and is not the difference between the two approaches. In fact, the authors themselves, in the supplement, use the same reference period for both approaches. I would thus suggest to not mention the "reference period" when discussing methodological differences with the AOU approach.

**Answer:**
We removed any reference to the "accounting for a reference period" from both the main text and the supplement. Given the numerical results from the previous point, we can conclude our approach and the one based on AOU are equivalent for practical purposes at the spatial and temporal scales considered in this study. We now only state this in the main text and supplement and we have removed the portion of text that states otherwise.

**Rev3:**
4. There is a recurring confusion between the condition of complete surface saturation ($SS_t = 1$ for all t at the surface, the assumption for AOU) with the condition of complete saturation initially everywhere ($SS_{t0} = 1$, which comes from the circular argument; point 1. above). This recurring confusion appears in both words and equations in the authors' response (to Reviewer 3, 2nd review, Major comment 2), in the main text (L270), and in the supplement (L102-103). This is strange because the assumption for AOU (surface saturation) is correctly stated elsewhere in the main text (L237) and even right after the incorrect assumption in the supplement, where the authors further suggest that they are the same (supplement L103-104):

(...) under the assumption that $SS_{t0} = 1$. This is in line with Duteil et al. (2013) that assumed saturation was reached at surface.

I would suggest removing any mention of this odd "initial saturation ($SS_{t0} = 1$)" condition which is not relevant to AOU.

**Answer:**
Contextually with all the other changes, we have removed any reference to "complete saturation at t0" from both the main text and the supplement.

**Minor points/suggestions**

**Rev3:**
L78:
*whilst away from the euphotic zone respiration exceeds primary production, resulting in net oxygen consumption.*
Is there any primary production away from the euphotic zone? If not, what about "whilst away from the euphotic zone respiration removes oxygen."

**Answer:**
We implemented the suggested change

**Rev3:**
L83: Replace "limit" with "reduce" for clarity

**Answer:**
We replaced "limit" with "reduce"

**Rev3:**
L270:
*The metrics in eq. 2 and 3 are related to the classic $O2 = O_{2,sat} - AOU$ decomposition, with the difference that, by explicitly accounting for a reference period, they relax the AOU assumption of complete saturation at t0.*
This is incorrect on two fronts: the difference does not lie in the reference time t0, and the AOU assumption is not complete saturation at t0 (see major point above).

**Answer:**
We removed here and from the rest of the text any reference to "complete saturation at t0" and "reference time" (see answer to major point).

**Rev3:**
Eq. (4): Note that "mix" as a subscript for the cross term could be confused for the contribution from (water) mixing.

**Answer:**
we replaced the subscript "mix" with "sord" (for Second ORDer).

**Rev3:**
L274: Add comma
*To assess what drives the oxygen changes we computed (...)*
like so
*To assess what drives the oxygen changes, we computed (...)*
otherwise it may read as "the oxygen changes (that) we computed".

**Answer:**
We added the commas.

**Rev3:**

L296: use minus signs instead of hyphens "−0.5 mg L$^{-1}$" (also in the exponent)

**Answer:**

done.

**Rev3:**

L302: missing minus sign in exponent in "6 mg L$^{-1}$

**Answer:**

done.

**Rev3:**

L327: missing space after "Fig 3d)"

**Answer:**

done.

**Rev3:**

L331: extra closing parenthesis in "(Fig. 4G-l))"

**Answer:**

done.

**Rev3:**

L333: missing "the" before "Central North Sea"

**Answer:**

done.

**Rev3:**

L375: missing space after "(Fig. 6d, e, f)."

**Answer:**

done.

**Rev3:**

Fig. 7 caption: Capitalize 1st letter of 2nd sentence.

**Answer:**

done

**Rev3:**

L408:

*(...) the correlation between SS and BResp is rather weak and, at some locations, positive (instead of negative as would be expected, Fig. 8e, f). This is because simple point-to-point correlation over the full period, does not allow to capture seasonally heterogeneous process.* Another reason could be that the point-to-point (in both time and space) correlation cannot capture the effect on SS from the distant respiration that occured in the past and at upstream locations from where the SS is computed.

**Answer:**

Our method may indeed fail when causes and effects are spatially decoupled, which we did discuss (L522-526). We don't believe this is the case here though: in IPSL and GFDL BResp decreases everywhere north of (and including in) the Central North Sea (Fig. 8 b,c). Due to the counterclockwise circulation any signal from distant respiration (or in this case lack thereof) in the Central North Sea would originate along the East coast of Britain, Shetland and Irish Shelf, where also BResp decreases. In the same regions SS does mostly increase, albeit little (Fig. 4 k,l). Hence we would still expect a negative correlation even if the signal came from upstream.

On the other hand we do present evidence that during winter months, when respiration is dominant over primary production, the correlation is negative as expected (Fig 8 h,i).

**Rev3:**

L450: replace comma after "etc" with dot

**Answer:**

done.

**Previous points where feedback was requested or required**

**Rev3:**

Sorry for this unclear comment:

Use words and function names in parentheses in the caption.

I assumed "nbias" and "nurmsd" were function names but maybe these are just acronyms? Anyway, what I intended to suggest was to use, e.g.:

*Fig. S1. Validation results. Plots show normalised bias (nbias) vs normalised unbiased root mean squared (nurmsd) for selected variables in the three ensemble members and in different model subdomains. A perfect fit would sit at the origin (0.0, 0.0).*

**Answer:**

We modified the caption as suggested.

**Rev3:**

Fig. 2 (formerly Fig. 3) comment:

Show past and future T and S too in appendix/supplement?

Sorry for not being explicit enough. I did not mean for the authors to move or remove this figure. Instead, I meant to suggest adding a supplement figure of past and future T and S to provide additional information and context to this ΔT and ΔS figure.

**Answer:**

We added figures to the supplement showing present and future temperature and ssalinity.

**Rev3:**
Fig. 3 (formerly Fig. 4) comment:
Show future O2?
As point above there I also wanted to suggest adding plots of future O2 without removing $\Delta$O2.

**Answer:**
We added figures in the supplementary material showing present and future $O_2$, $O_{2,sat}$ and SS.